# GENERATING SAMPLES TO PROBE TRAINED MODELS

## ABSTRACT

There is a growing need for investigating how machine learning models operate. With this work, we aim to understand trained machine learning models by questioning their data preferences. We propose a mathematical framework that allows us to probe trained models and identify their preferred samples in various scenarios including prediction-risky, parameter-sensitive, or model-contrastive samples. To showcase our framework, we pose these queries to a range of models trained on a range of classification and regression tasks, and receive answers in the form of generated data.

## 1 INTRODUCTION

Machine learning models are widely used in today's data-driven world, powering critical decision-making processes in sectors ranging from healthcare to human resources. Their widespread adoption in high-stakes scenarios raises important questions on aligning trained models with human values. Understanding how these models operate has become a critical concern. Our quest along this line starts with the following inquiry: *What kind of data can we generate to probe our trained models?*

To respond to this inquiry, we study the implicit data distribution favored by trained models. In other words, our approach to understanding a model is based on creating samples in the data domain that the trained model considers favorable for a specific task. Unlike conventional ML pipelines that focus on static datasets and predictive accuracy, our approach enables dynamic interrogation of model behavior via investigating the answers of the model to specific questions. We show that these questions can be customized to each situation and they can be expressed mathematically through a loss function that evaluates the data based on a combination of data characteristics and model parameters. We consider the problem of understanding a model to be a more nuanced endeavor that requires exploration across multiple dimensions of questioning. This involves providing explanations, such as counterfactual (Wachter et al., 2017) or prototypical (Biehl et al., 2016) scenarios, shedding light not only on why a particular prediction was made, but going beyond it as well. For instance, insights into model behavior can be gained by generating parameter-sensitive data samples. When two models showing similar performances give different predictions, also known as predictive multiplicity (Marx et al., 2020), our approach can be used to generate data to systematically compare their behaviors in diverse scenarios. These custom questions, and others, provide a qualitative understanding of the model. In addition, users have the flexibility to customize queries by designing specific probing functions within the data space.

**Related Literature.** Our work complements extensive research in synthetic data generation that has been pivotal in addressing fairness, bias detection (Kusner et al., 2017) and reduction (Xu et al., 2018; van Breugel et al., 2021) as well as dataset augmentation (Wong et al., 2016; Fawaz et al., 2018). Generative Adversarial Networks (GANs) and Variational Autoencoders (VAEs) have been widely used to approximate data distributions (Goodfellow et al., 2014; Xu et al., 2018; Kingma & Welling, 2014; Breugel et al., 2024), focusing on privacy, diversity, and fidelity as primary goals.

Recent studies leveraged generative models for counterfactual generation and exploring underrepresented data regions. For example, Joshi et al. (2019) proposed a framework for generating task-specific synthetic data, enhancing model explainability. Similarly, Redelmeier et al. (2024) introduced an approach using autoregressive generative models to create counterfactuals, facilitating bias exploration and decision boundary analysis. Recent work on global counterfactual explanations has further expanded the scope of interpretability by targeting group-level understanding. Rawal & Lakkaraju (2020) introduced a framework for generating global, rule-based recourse summaries for

population subgroups, optimizing objectives like accuracy, coverage, and cost. These summaries are 'if-then' rules linking subgroup characteristics to actionable feature changes that influence a model's prediction. Plumb et al. (2020) proposed interpreting clusters in low-dimensional representations by finding sparse transformations that align one group with another.

Energy-based models (EBMs) have also emerged as a promising framework, combining generative and discriminative modeling tasks. By treating classifier logits as an energy function, EBMs can model joint distributions over data and labels (LeCun et al., 2006; Duvenaud et al., 2020). Applications of EBMs include adversarial robustness, out-of-distribution detection, and data augmentation (Zhao et al., 2017; Liu et al., 2020; Arbel et al., 2021; Margeloiu et al., 2024). For instance, Duvenaud et al. (2020) demonstrated improved out-of-distribution detection using a joint energy-based model, while Ma et al. (2024) extended EBMs to tabular data for synthetic data generation.

The proposed framework draws inspiration from these works while introducing a distinct perspective. Our probing function can be seen as an energy function and leads to Gibbs distribution. However, rather than learning the energy function to capture the data distribution (conditioned on label), we create a probing function using trained models. This design allows the distribution to generate samples that address the specific posed question. Related works, such as (Duvenaud et al., 2020) and (Ma et al., 2024), adopt a similar approach by utilizing a trained classifier to obtain an energy function and using Langevin dynamics for sampling from the Gibbs distribution. However, their main objective is to mimic the true data distribution. In fact, the former paper combines training of the energy function and classifier. In contrast, we propose a flexible framework that allows for directing diverse queries to trained models via probing functions that reflect various objectives, such as identifying prediction-risky, parameter-sensitive, or model-contrastive data samples.

**Contributions.** We introduce a new inductive approach that generates data samples through a flexible probing function designed to analyze and reveal the behavior of a trained model. Our method can be tailored to suit various classification and regression tasks, demonstrating its versatility in producing data that meet specific queries. This work serves as a foundational step in establishing the effectiveness and potential of our approach.

## 2 THE MATHEMATICAL FRAMEWORK

First, we present our notation. The labeled data lie in $\mathcal{X} \times \mathcal{Y}$, and the model defines a predictor function $f(\boldsymbol{\theta}, \cdot) : \mathcal{X} \to \mathcal{Y}'$ for any given set of model parameters $\boldsymbol{\theta} \in \Theta$. For a given sample $\mathbf{x} \in \mathcal{X}$, the predicted label $y_{\boldsymbol{\theta}}(\mathbf{x}) \in \mathcal{Y}$ is obtained from the predictor function. The cost function $\ell_F : \mathcal{Y} \times \mathcal{Y} \to \mathbb{R}_{\geq 0}$ measures how far the predicted labels are from the true labels.

The standard construction of the parameter loss function is

$$F(\boldsymbol{\theta}) = \int_{\mathcal{X} \times \mathcal{Y}} (\ell_F(y_{\boldsymbol{\theta}}(\mathbf{x}), y) + R_F(\boldsymbol{\theta})) \mathrm{d}\nu(\mathbf{x}, y) = \frac{1}{N} \sum_{i=1}^{N} \ell_F(y_{\boldsymbol{\theta}}(\mathbf{x}_i), y_i) + R_F(\boldsymbol{\theta}), \quad (1)$$

which can be seen as an integral of $\ell_F + R_F$ against the empirical distribution given by the training dataset $\{(\mathbf{x}_i, y_i)\}_{i=1}^{N} \subseteq \mathcal{X} \times \mathcal{Y}$. Here, $R_F(\boldsymbol{\theta})$ is a regularizer term that depends only on $\boldsymbol{\theta}$.

Figure 1 provides an overview of our framework. Just as the training process, which uses $F$ to find the right parameters, our framework probes the model with a function $G$ defined on the data space. In the variational setting, the symmetry is clear, where we get a distribution over the parameters (data) instead of a single $\boldsymbol{\theta}^*$ ($\mathbf{x}^*$). The loss function $F$ in equation 1 is an average over observed data, and, similarly, we construct the data loss $G$ by integrating out $\boldsymbol{\theta}$ from a curated function (described later) that depends both on data and model parameters. We design this function of data and parameters to attain low values when the desiderata of our probing scenario are met. Different choices correspond to posing different questions to the model, and allow us to gain valuable insights into its behavior by observing the generated data, both qualitatively and through population-level statistics.

The blue arrows (a) and (c) in Figure 1 map (loss) functions on the respective spaces $\Theta$ (parameter space) and $\mathcal{X}$ (data space) to distributions over the same spaces. This corresponds to solving the Bayesian Learning Problem (BLP), which –in the case of (a)– is

$$\underset{q \in \mathcal{Q}}{\arg\min} \, \mathbb{E}_q[F] - \tau \mathcal{H}(q), \quad (2)$$

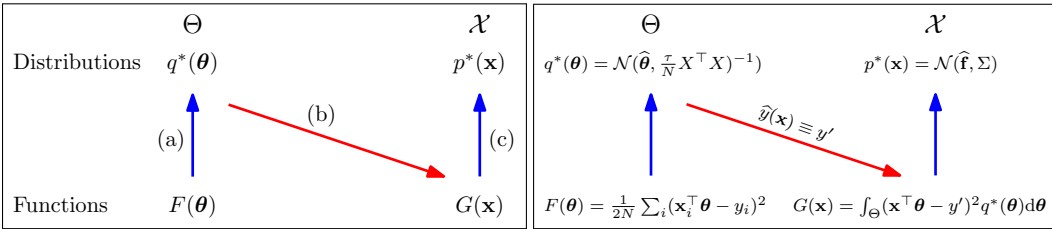

Figure 1: (Left) Overview of model probing by data generation. Samples from $p^*(\mathbf{x})$ answer the question posed by $G$. The vertical arrows (a) and (c) start with functions and lead to distributions on the same space by solving equation 2 and equation 3. The diagonal arrow (b) starts with a distribution on the parameter space and obtains a loss function on the data space by integrating out $\boldsymbol{\theta}$ dependence of a function on $\Theta \times \mathcal{X}$ against the distribution $q^*(\boldsymbol{\theta})$. (Right) The special case of the Linear Regression (LR) model with mean square error admits an analytic solution. The $G$ function is designed to find data points $\mathbf{x}$ whose solutions under LR are close to a chosen prediction $y'$ and averaged over $q^*$. The distribution $p^*(\mathbf{x})$ is calculated to be a Gaussian distribution centered at a point $\hat{\mathbf{f}}$, which is shifted from the mean of given data by a certain amount depending on the desired output value $y'$. Explicit forms of $\hat{\mathbf{f}}, \Sigma$ and $\widehat{\boldsymbol{\theta}}$ and their derivation can be seen in Appendix A.

where $\mathcal{Q}$ is a choice of candidate distributions on $\Theta$, and $\mathcal{H}(q) = -\int_\Theta q \log q \, \mathrm{d}\mu$ is the entropy with respect to a base measure $\mu$. The problem can be interpreted as an implementation of the exploration-exploitation trade-off in the parameter space. The constant $\tau > 0$ is called the temperature and balances these two objectives. If $\mathcal{Q}$ is the set of all density functions, then the Gibbs-Boltzmann distribution $q^*(\boldsymbol{\theta}) \propto e^{-\frac{1}{\tau}F(\boldsymbol{\theta})}$ is the unique solution to equation 2.

Symmetrically on the data space $\mathcal{X}$, the blue arrow labeled (c) in Figure 1 represents solving

$$\arg\min_{p \in \mathcal{P}} \mathbb{E}_p[G] - \tau \mathcal{H}(p). \tag{3}$$

The distribution $p^*(\mathbf{x}) \propto e^{-\frac{1}{\tau}G(\mathbf{x})}$ is its global solution, balancing the expectation term's effect of mass concentration at low $G$-values, with the entropy term's effect of exploring the data space.

There are various methods of sampling from equation 3. In this work, we used Metropolis Adjusted Langevin Algorithm (MALA) to sample directly from the Gibbs-Boltzmann distribution $p^*(\mathbf{x}) \propto e^{-\frac{1}{\tau}G(\mathbf{x})}$. This method is a kind of noisy gradient descent, with an acceptance/rejection step ensuring that the limiting distribution is $e^{-\frac{1}{\tau}G(\mathbf{x})}$. Details of this method are given in Appendix C. Alternatively restricting the problem to a statistical manifold $\mathcal{P}$, Variational Inference (VI) can be used to effectively reach a distribution $p^* \in \mathcal{P}$; see (Ganguly & Earp, 2021; Geiser, 2020). For example, if $\mathcal{P}$ were chosen to be the Gaussians, then one would only need to keep track of the mean and the covariance in learning a $p^* \in \mathcal{P}$.

The red arrow (b) constructs the function $G$ as an integral of a function over both the data and parameter spaces, obtained by integrating out $\boldsymbol{\theta}$ with respect to the measure $q^*(\boldsymbol{\theta})$, in direct analogy with the construction of $F$ in equation 1. This construction of $F$ is an integral against the empirical data distribution and, therefore, the learned parameters are compatible with the training data. Analogously, constructing $G$ ensures that the search over $\mathcal{X}$ remains compatible with parameters sampled from $q^*(\boldsymbol{\theta})$. In particular, if $q^*(\boldsymbol{\theta})$ is a is a Dirac-delta distribution, this reduces to a single parameter vector $\boldsymbol{\theta}^* \in \Theta$. The specific choice of the integrand for $G$ determines which samples from $p^*(\mathbf{x})$ are the data points that answer a question posed about the trained model.

Lastly, note that we can change our search space by replacing $G$ with $G \circ \varphi$ for some $\varphi : \mathcal{Z} \to \mathcal{X}$. In high-dimensional data spaces $\mathcal{X}$, we will use this setup with $\mathcal{Z}$ as the latent space and $\varphi$ as the decoder function of a pre-trained Variational AutoEncoder (VAE). In this case, $p^*(\mathbf{z})$ becomes a distribution on $\mathcal{Z}$, and mapping its samples to $\mathcal{X}$ by $\varphi$, gives points on the data manifold. Using a pre-trained VAE for image models reduces the data space dimension, and therefore, the efficiency of the MALA sampling process, but there are also conceptual benefits. There are vast regions of the high-dimensional input space that do not correspond to plausible images and which the model did not encounter during the training process. Some of these regions may also satisfy our query $G$. Therefore, using a pre-trained VAE decoder reflects a plausible image requirement, if imposed.

## 3 PROBING TRAINED MODELS

We start with a general structure of the loss function curated for questioning trained models:

$$G(\mathbf{x}) = \int_{\Theta} \ell_G(y_{\boldsymbol{\theta}}(\mathbf{x}), \widehat{y}(\mathbf{x})) q^*(\boldsymbol{\theta}) \mathrm{d}\boldsymbol{\theta} + R_G(\mathbf{x}), \tag{4}$$

where $\widehat{y}$ stands for a predictor and $R_G$ is a regularizer function that can be chosen to put additional soft constraints on the samples in addition to the hard constraints coming from the restriction $p \in \mathcal{P}$. This general probing function enables us to express a wide range of model-inspection tasks. In the remainder of this section, we walk through several representative cases, each capturing a specific type of question one might ask about a trained model's behavior. These are not exhaustive, but illustrative scenarios that demonstrate the flexibility of the proposed framework. Figure 2 showcases what sort of data points would be produced in a synthetic dataset.

In many cases we pick the regularizer term as $R_G(\mathbf{x}) = \lambda \|\mathbf{x} - \mathbf{x}_a\|_r^r$ for $r \geq 1$, so that the generated synthetic data is localized to an anchor $\mathbf{x}_a$. The $\lambda > 0$ term is an explicit tunable parameter that controls the trade-off between the probing objective and localization strength. In fact, different weightings can also be applied to different columns to enforce this more or less stringently for different features.

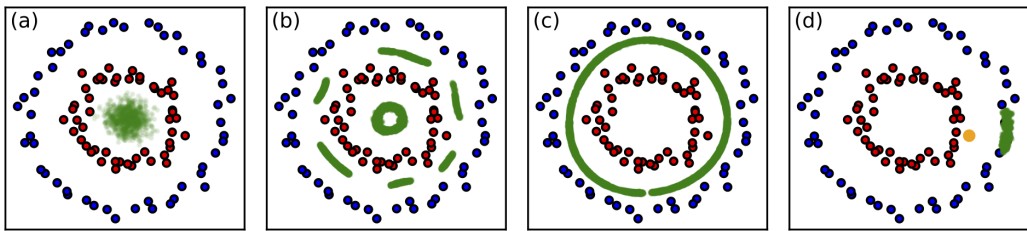

Figure 2: Given a dataset of two concentric circles labeled red and blue, two Support Vector Machine (SVM) models are trained on the binary classification task with kernels chosen as Radial Basis Function (RBF) and cubic polynomial, respectively. The generated data points are green. In (a), we contrast the two SVM models, looking for samples for which their predictions differ, and discover that this is the case in a region near the origin lacking any training points. In (b) and (c), we inquire about data points that would be considered risky by the two models using RBF and cubic kernels, respectively. In (d) we design $G$ so that it generates data points which are classified with the opposite label of the orange point by the RBF-SVM without straying too far from it.

**Model-contrasting samples.** Given two models, finding data points where their predictions disagree is illuminating either to interpret model-specific biases or to audit consistency between two different models. This is particularly useful when comparing models with different inductive biases (MLP vs. CNN, linear vs. nonlinear, and so on). Given $\widehat{y}$ as the predictor functions of the model that is being compared against (which can be non-parametric, like in boosted trees), we solve equation 3 with the function

$$G(\mathbf{x}) = \ell_G(y_{\boldsymbol{\theta}^*}(\mathbf{x}), 1 - \widehat{y}(\mathbf{x})) + R_G(\mathbf{x}). \tag{5}$$

In Figure 2(a), we contrast two SVM models with different kernels and discover a region near the origin, which does not contain any samples from the dataset but would give conflicting predictions if a new sample were to come from it.

**Prediction-risky samples.** To identify the indecisive regions in a model's decision surface, we either put $G(\mathbf{x}) = \|f(\mathbf{x}, \boldsymbol{\theta}^*) - \alpha\|_r^r$ for $r \geq 1$, where $f$ may be the decision function of a binary classification model such as in SVM and logistic regression and $\alpha$ denotes the cutoff point, or given a multi-class classification where $f$ are the prediction probabilities, we put $G(\mathbf{x}) = -H(f(\mathbf{x}))$, the negative entropy. Solving equation 3 thus corresponds to generating "risky data points" near the decision boundary, yielding insights into whether the model correctly identifies important aspects of the data for decision-making and whether that aligns with the user's conceptions. As an illustrative example in Figure 2(b), we search for the decision boundary of a RBF-SVM trained on a dataset of two concentric circles, and we discover the ring in the center as an unexpected decision boundary.

**Parameter-sensitive samples.** Given a set of parameters $\boldsymbol{\theta}^*$ and a distribution $q^*(\boldsymbol{\theta})$ of parameter values, we ask the model for data samples whose classifications would flip if the model parameters were to be (perhaps slightly) perturbed. This can be achieved by solving equation 3 using

$$G(\mathbf{x}) = \int_\Theta \ell_G(y_{\boldsymbol{\theta}}(\mathbf{x}), 1 - y_{\boldsymbol{\theta}^*}(\mathbf{x})) q^*(\boldsymbol{\theta}) \mathrm{d}\,\boldsymbol{\theta} + R_G(\mathbf{x}). \tag{6}$$

This integral would be approximated by samples from $q^*$. When $q^* \in \mathcal{Q}$ is chosen from a restricted family of distributions, like Gaussians with fixed variance, sampling from $q^*$ means perturbing $\boldsymbol{\theta}^*$. This probing function is particularly useful when we want to examine the consistency of a model's predictions under small shifts in its parameters. By identifying inputs whose predictions vary significantly with minor parameter changes, we can highlight sensitive regions in the input space—areas that might indicate over-dependence on specific parameter configurations. This can have overlaps but is distinct from prediction-risky samples, as we showcase in our computational study section. Parameter-sensitive samples has the flexibility to be generated far from the decision boundary, especially in non-linear models.

In equation 6 we formulated the probing function for the binary case. For regression models, choosing $G(\mathbf{x}) = \int_\Theta \exp\left(-\|y_{\boldsymbol{\theta}}(\mathbf{x}) - y_{\boldsymbol{\theta}^*}(\mathbf{x})\|^2/\sigma^2\right) q^*(\boldsymbol{\theta}) \mathrm{d}\,\boldsymbol{\theta} + R_G(\mathbf{x})$ has lower value when the predictions are large (large being measured with a yardstick of size $\sigma$ of our choosing). Similarly for equation 5. As for multi-class prediction we can also use this $G$, with $y_{\boldsymbol{\theta}}(\mathbf{x})$ representing the (post-softmax) probabilities of class predictions.

**Fixed-label samples.** Finally, we probe the model for what it thinks are good data samples that fit the bill for the prediction $y'$, either for a single parameter $\boldsymbol{\theta}^*$ or a distribution $q^*(\boldsymbol{\theta})$

$$G(\mathbf{x}) = \ell_G(y_{\boldsymbol{\theta}^*}(\mathbf{x}), y') + R_G(\mathbf{x}) \quad \text{and} \quad G(\mathbf{x}) = \int_\Theta \ell_G(y_{\boldsymbol{\theta}}(\mathbf{x}), y') q^*(\boldsymbol{\theta}) \mathrm{d}\,\boldsymbol{\theta} + R_G(\mathbf{x}), \tag{7}$$

respectively. Here, $R_G(\mathbf{x})$ is a localizer at an anchor point. We can take a data point $(\mathbf{x}_0.y_0)$ to be this anchor. In case $y' \neq y_0$, we are exploring changes in $\mathbf{x}_0$ that would need to happen for the prediction to change; in other words, a counterfactual; see Figure 2(d). In case $y' = y_0$ with a weak localizer, we can obtain a sample that would lead to a similar prediction, *i.e.*, a factual.

Figure 1 demonstrates the steps when $y_{\boldsymbol{\theta}}(\mathbf{x}) = \mathbf{x}^\top \boldsymbol{\theta}$ corresponds to linear regression, and both $\ell_F$ and $\ell_G$ are the mean squared errors. For this special case, we obtain analytical solutions for all steps of our framework. The details of this observation are given in Appendix A.

**Feature-restricted samples.** By restricting $\mathcal{P}$ to be supported on data with certain features fixed, such as those features corresponding to age, race, and so on, we can ask the model for all of the above questions but conditioning on certain immutable characteristics. This falls into the class of optimizations, where instead of $G$ we consider $G(\varphi(\boldsymbol{z}))$ on some other (latent) space $\boldsymbol{z} \in Z$. In case of image data, for example, to have our samples conform to the data manifold, $\varphi$ can be taken as the trained decoder module from a VAE. Pushforwards $\varphi_* \tilde{p}$ of measures $\tilde{p} \in \mathcal{P}(Z)$ on the latent space then lie on the data manifold, *i.e.*, sampling $\boldsymbol{z} \sim \tilde{p}$ and computing $\varphi(\boldsymbol{z})$ gives a data sample. See Figure 4 for this method in a concrete application.

## 4 COMPUTATIONAL STUDY

In this section, we conduct a series of experiments to evaluate the cases presented in Section 3. Our experiments aim to evaluate the proposed framework by demonstrating its ability to generate data samples across various scenarios. We use well-established datasets that have been recently adopted in related literature (e.g., (Good et al., 2023), (Ley et al., 2023), (Si et al., 2024)), and their specifics are outlined in Appendix B. The implementation details and code for reproducing these experiments are available on our GitHub repository.[1]

**Model-contrasting samples.** This experiment investigates the differences between two predictive models by probing the features that drive contrasting predictions for the same data. Through our framework, we pose the following question:

*Which features or input changes lead to disagreement between the two models' predictions?*

---

[1]https://anonymous.4open.science/r/EvD-6FB1/

To explore this, we apply the framework to datasets of different modalities. For tabular data, we use the FICO dataset (FICO, 2018); for image dataset, we use MNIST (LeCun et al., 2010).

We begin by investigating model divergence in scenarios where the comparison model is non-differentiable. To this end, we train XGBoost Chen & Guestrin (2016) -a non-parametric model-alongside logistic regression on the FICO dataset, which consists of credit applications with features related to financial history and risk performance. This setup highlights the flexibility of our framework, as it enables probing differences between models with fundamentally distinct modeling approaches. Although the two models agree on 94.5% of the predictions in the test set, our framework generates a set of samples where their predictions exhibit full disagreement, *i.e.*, XGBoost predicts one class, while logistic regression predicts the opposite. Figure 3 presents the feature distributions for these discrepant samples, focusing on three representative features. We observe that disagreement tends to occur when the number of credit accounts opened in the past year falls in the 12–15 range, which is outside of this feature's distribution in the test data. This indicates that our method can surface disagreement patterns that would remain undetected through standard evaluation alone. We also include in Appendix D a comparison between a linear model and a Support Vector Regression (SVR) model using a different tabular dataset.

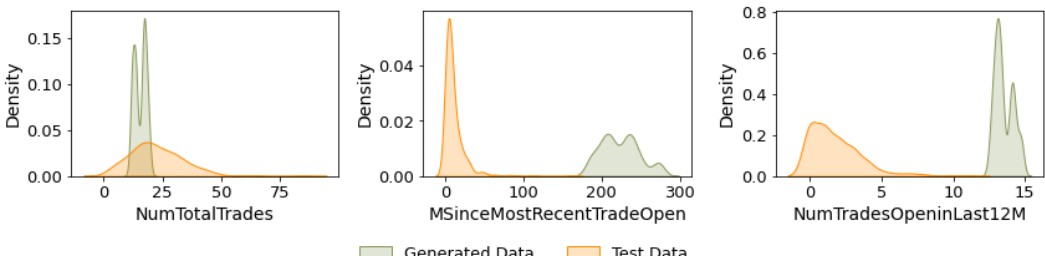

Figure 3: The distributions of three representative features in the generated samples. Here, XGBoost predicts "Bad" for `RiskPerformance`, while logistic regression predicts "Good".

Our framework can also be used to compare and contrast two models trained on image data. To demonstrate, we consider a Convolutional Neural Network (CNN) and an MLP, both trained on MNIST. The architectures of these networks are provided in Appendix B.2. To better capture the data manifold, we also train a VAE with a latent dimension of 10. The trained encoder module of the VAE is denoted by $z \mapsto \varphi(z)$. Further details on the VAE training process are provided in Appendix E. In Figure 4, we present an example computation illustrating how this setup works. Starting with a latent vector encoding an image with label '3', we sample from a distribution that prefers the label '8' jointly for both a trained CNN (LeNet5) and an MLP.

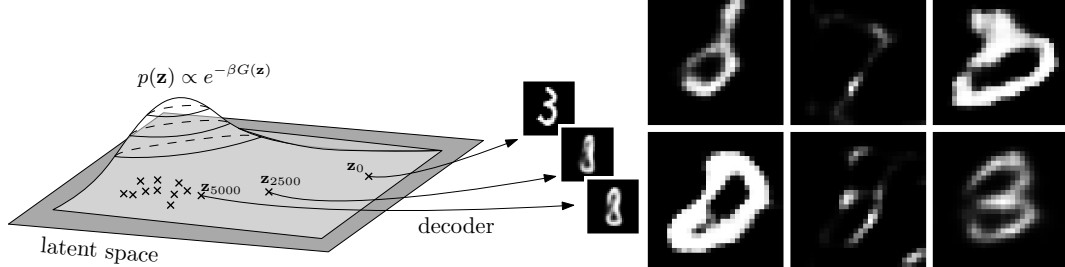

Figure 4: (left) Using Langevin dynamics in the latent space, we obtain a sequence of latent vectors that, when passed through the decoder $\varphi$, correspond to a walk on the data manifold. In this image, the function $G$ is the sum of cross-entropy predictions of trained MLP and LeNet5 networks for the label '8' and for the data $\varphi(z)$. (right) Images in the first and second columns are generated to prefer a given label on an MLP model and another one on a CNN model. upper-left: CNN-'0' MLP-'1', upper-middle: CNN-'1' MLP-'7', lower-left: CNN-'0' MLP-'8', lower-middle: CNN-'2' MLP-'5'. On the third column, the upper image prefers the label '8' for the MLP model whilst being close to a data sample with label '3', and the same for the lower image for the CNN model.

We use this setup to systematically compare the CNN and MLP models. In Figure 4, we showcase some samples generated by forcing functions $G$ that pull the data toward incompatible directions, for example, resulting in amorphous data points that exhibit characteristics of both '1' and '0'. The third column highlights cases where the label '8' is preferred (top: MLP, bottom: CNN) while remaining close to an actual MNIST image labeled '3', which is enforced through two-norm regularization.

**Prediction-risky samples.** As our first example, we train an MLP on the FICO dataset to classify credit risks as "Good" or "Bad." Prediction-risky samples are those for which the model outputs softmax probabilities close to 0.5, reflecting high uncertainty. A detailed analysis of this experiment, along with the generated samples, is provided in Appendix D. For example, we see in Figure 12 that the model assesses between three-to-five delinquencies (MaxDelqEver) as being on the edge between being a bad credit risk versus a good credit risk. A domain expert can then assess if this coincides with her expectations of the model.

As another example, in order to demonstrate the versatility of our approach, we apply it to tree-based models. Using these models poses a challenge due to the locally constant nature of their prediction functions, and hence, one cannot directly use the gradient-based methods. However, in Appendix C.2, we describe how approximate gradient information can still be leveraged effectively to overcome this limitation.

Using the wine dataset from scikit-learn, we train a Random Forest (RF) classifier. This dataset is designed for classification tasks and consists of 13 numerical features that describe various characteristics of wine, such as hue and alcohol content. The target variable represents the wine's region of origin, which falls into one of three distinct classes. We ask the following question:

*Which input samples drive the RF classifier to produce nearly uniform class probabilities?*

To highlight the flexibility of our method, we also impose a regularizer that encourages that a Decision Tree (DT) fitted to the same dataset predicts a given region with certainty. This can be achieved by letting $G(\mathbf{x})$ to simultaneously maximize the entropy of the RF's prediction probabilities (encouraging uncertainty) and minimize the distance between the DT's prediction probabilities and a fixed one-hot vector, thereby enforcing certainty on a chosen class. Both models have high accuracy on the validation set (RF: $94.4\%$, DT: $88.8\%$). Therefore, the generated data's features necessarily lie outside the empirical data distribution. We generated 50 data points such that the DT predicts class_1 with full certainity, and the RF's prediction probabillites are $(0.31_{\pm 0.03}, 0.4_{\pm 0.06}, 0.29_{\pm 0.06})$. See Table 1 in Appendix D to compare the feature values of this generated wine feature dataset versus those from each of the three regions. Following the decision path of the DT, we observe that the generated samples are identified as belonging to class_1 solely based on their color intensity. In contrast, the Langevin process resulted in a set of wine features for which the model exhibited uncertainty the random forest classifications.

**Parameter-sensitive samples.** This experiment investigates data samples that are sensitive to small perturbations in the model parameters. Unlike prediction-risky samples, parameter-sensitive samples may exist anywhere in the input space, as their classification changes with slight shifts in the model's parameters. To guide this analysis, we pose the following question:

*What kind of data samples vary in classification due to small changes in model parameters?*

We train an MLP on the FICO dataset and generate parameter-sensitive samples by perturbing the model parameters using a Gaussian distribution centered at the original weights with fixed variance. Using the probing function in (6), we generate and analyze 500 such samples to identify instances most susceptible to model variation, and compare them with prediction-risky samples.

Figure 5 shows density plots of four representative features (see Appendix D for additional ones), comparing parameter-sensitive and prediction-risky samples. By comparing these two distributions, we gain insights into how the model perceives uncertainty from different perspectives. While the prediction-risky samples are associated with uncertainty near the decision boundary, the parameter-sensitive samples highlight regions in the feature space where small parameter changes can flip predictions. The features AverageMinFile (average observation period) and NumTotalTrades (total number of trades) exhibit similar distributions across both sets. In contrast, the features MSinceMostRecentTradeOpen (months since most recent trade) and NumInqLast6M (inquiries in the past six months) diverge. For example, NumInqLast6M, which signals recent credit-

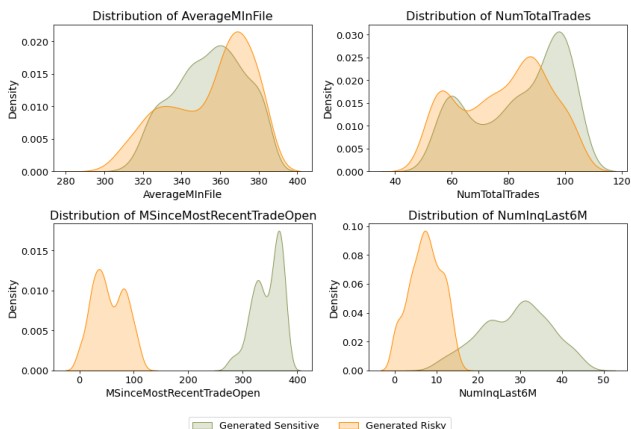

Figure 5: Feature distributions in generated parameter-sensitive and prediction-risky samples.

seeking activity, is lower among prediction-risky samples, indicating that individuals with fewer recent inquiries are more likely to fall near the decision boundary. In contrast, parameter-sensitive samples exhibit a broader distribution, indicating that parameter shifts affect individuals across a wider range of credit inquiry patterns. This may be because frequent inquiries reflect diverse financial behaviors, making these samples more vulnerable to prediction instability. These findings suggest that some features contribute more to robustness under parameter variation, while others primarily influence boundary-sensitive classifications.

**Fixed-label samples.** We apply the probing function $G$ in equation 7 to the Adult dataset (Becker & Kohavi, 1996), a widely used benchmark for binary classification based on income level (whether an individual earns more than $50K annually). A logistic regression model is trained on the dataset, and we examine its behavior by constructing counterfactual samples. Specifically, given a factual instance $(\mathbf{x}_0, y)$, we generate samples using the probing function with $y' \neq y$ and regularizer $R(\mathbf{x}) = \|\mathbf{x} - \mathbf{x}_0\|^2$. In this experiment, the factual instance represents a Latin-American Black Female, predicted to earn less than $50K. Using our framework, we pose the following question:

*What feature changes would cause the model to predict an income above $50K for this individual?*

To address this question, the probing function is designed to balance two objectives: steering the model prediction toward the target label $y' = 1$ (using cross-entropy loss), and staying close to the original input (via the regularizer term $R_G(\mathbf{x})$).

Figure 6 shows the distribution of the generated samples aggregated over 50 independent runs. The shaded regions (for numerical features) and error bars (for categorical features) indicate variability across runs. The results provide insights into the model's classification process and the factors it deems influential in income predictions. While generating counterfactual samples, we impose bounds on age, educational attainment, and weekly working hours, which are enforced during Langevin dynamics sampling process by clipping each step to remain within the specified ranges. All of the of the generated samples are predicted to have label $y = 1$ (income above $50k). Comparing the factual input with the counterfactual distribution reveals significant categorical shifts. For example, the majority of samples indicate a change in gender from female to male, and a region shift from Latin America to Western Europe, suggesting that these features significantly influence the model's decision. These observations raise questions about fairness and bias. While we may directly investigate the logistic regression coefficients associated with these features, *e.g.,* `female` $\approx -1.375$, `male` $\approx -1.243$, the bias is more clearly revealed through the generated samples. More importantly, such coefficients are not readily available for more complex models like deep networks Figure 6 shows a roughly equal distribution for the gender feature. Given that our probing function also includes a proximity term to the initial data point who is a female, we deduce that gender is influential for the model's prediction, otherwise we would expect the generated counterfactuals to remain female due to the localizer term in $G$.

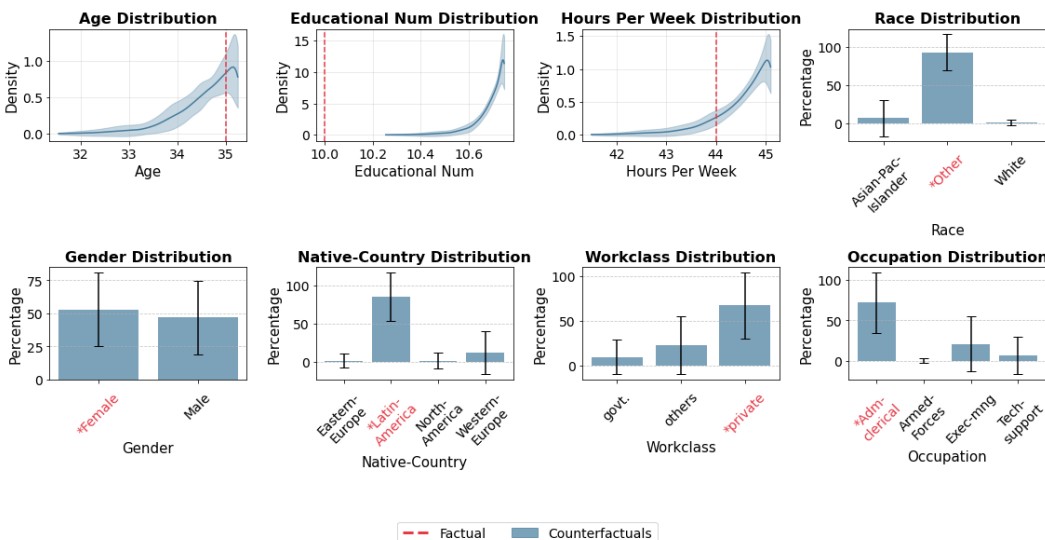

Figure 6: Feature distributions of generated counterfactual samples (blue shaded) with factual instance highlighted (red markers).

In addition to the main experimental results, we conduct a comparison with DiCE (Mothilal et al., 2020), a widely used counterfactual generation method. DiCE produces a set of diverse counterfactual instances by optimizing for feature changes that flip the model's prediction. In contrast, our probing scenario constructs a full stationary distribution over inputs aligned with the model's behavior, rather than a finite collection of point solutions. This yields a richer characterization of how the model responds to perturbations and provides a distributional view of its decision boundary. The comparison results are provided in Appendix D. Consistent with the design of our probing scenario, all generated samples strictly satisfy the decision-flipping condition. In other words, every point in the distribution changes the classifier's prediction relative to the factual instance. The results illustrate that our method and DiCE are complementary. While DiCE offers example-level counterfactuals, our approach captures the underlying probability landscape from which such counterfactuals arise.

**Generating samples in high dimensions.** We probe the ResNet50 model (pre-trained on ImageNet-1k dataset, available in `torchvision.models`) to demonstrate that our approach scales to high-dimensional data spaces ($224 \times 224 \approx 50$k). By generating images at successively higher resolutions and using the pretrained TAESD autoencoder (Bohan, 2023), we obtain images guided by a probing function $G$ that favors latent vectors $z$ such that the decoded images have low reconstruction loss under the VAE and are predicted by ResNet50 to be "goldfish" or "snail", respectively (Figure 7 in Appendix F). The generated "goldfish" images predominantly feature orange regions with a small black dot resembling an eye, while the "snail" images include curved antenna-like shapes. This suggests that ResNet50 relies strongly on color cues for goldfish detection.

We test this hypothesis by evaluating ResNet50 on 50 color-modified validation samples. Results for goldfish, snail, and other classes are reported in Table 2 in Appendix F. Notably, swapping green and blue channels does not substantially impair goldfish detection, while grayscale conversion sometimes has only a limited effect on classification.

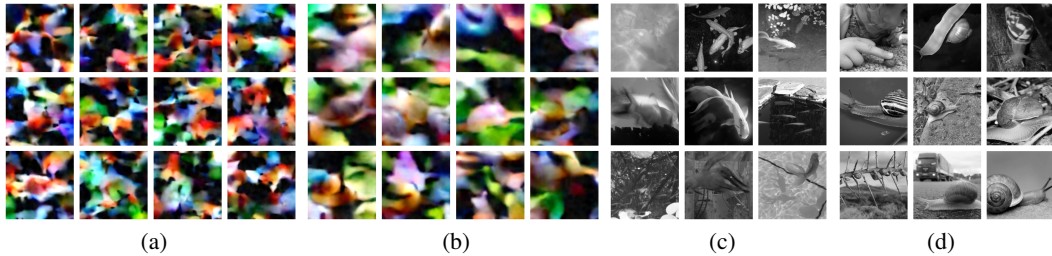

(a)        (b)        (c)        (d)

Figure 7: (a) generated samples classified as "goldfish", (b) generated samples classified as "snail", (c) "goldfish" validation images misclassified by ResNet50 when converted to grayscale, (d) "snail" validation images converted to grayscale, only the first is misclassified by ResNet50.

## 5 CONCLUSION

We introduce a mathematical framework for probing trained models with tailored data samples designed to answer specific queries, going beyond traditional interpretability methods. By formulating probing functions, we demonstrate how to generate samples for scenarios like prediction risky, parameter sensitivity, and model contrast. Our computational study shows the framework's effectiveness in classification and regression tasks on diverse datasets, revealing insights into decision boundaries and input sensitivities. Our goal is to understand machine learning models for positive societal impact, with our tools supporting model analysis through sample generation.

Our framework offers opportunities for improvement and future research. Incorporating implicit constraints among features (*e.g.*, monotonic relationships) could enable the generation of samples that accurately represent the dataset and enhance their interpretability and reliability. Applying our framework in various application areas with domain experts could also illuminate different usability aspects. Addressing these considerations will help refine and build upon the foundational study presented here.

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

## A    LINEAR REGRESSION WITH GAUSSIAN DATA

We start with $y_{\boldsymbol{\theta}}(\mathbf{x}) = \boldsymbol{\theta}^\top \mathbf{x}$ and $\ell_F(y, y') = \ell_G(y, y') = \frac{1}{2}(y - y')^2$. Given a dataset $\{(\mathbf{x}_i, y_i)\}_{i=1}^N$, we construct the loss function $F(\theta)$ as the integral of $\ell_F(y_{\boldsymbol{\theta}}(\mathbf{x}), y)$, over the data distribution, which is approximated by the Dirac delta comb $\nu = \frac{1}{N} \sum_{i=1}^N \delta_{(\mathbf{x}_i, y_i)}$:

$$F(\theta) = \int_{\mathcal{X} \times \mathcal{Y}} \ell_F(y_{\boldsymbol{\theta}}(\mathbf{x}), y) \mathrm{d}\,\nu(\mathbf{x}, y) = \frac{1}{2N} \sum_{i=1}^N |\mathbf{x}_i^\top \theta - y_i|^2.$$

Assume, for convenience, that a constant feature of 1 is included as the last coordinate of $\mathbf{x}$, allowing us to explicitly represent the intercept. Using this notation, we define

$$\mathbf{x} = \begin{bmatrix} \mathbf{f} \\ 1 \end{bmatrix}, \qquad \theta = [\boldsymbol{\xi} \quad b], \text{ so that } \mathbf{x}^\top \theta = \mathbf{f}^\top \boldsymbol{\xi} + b.$$

We write the design matrix as

$$D = \begin{bmatrix} \cdots & \mathbf{x}_1^\top & \cdots & 1 \\ \cdots & \mathbf{x}_2^\top & \cdots & 1 \\ & \vdots & & \vdots \\ \cdots & \mathbf{x}_N^\top & \cdots & 1 \end{bmatrix} = [\; X \quad \mathbf{1} \;].$$

The quadratic loss function can then be expressed as

$$F(\theta) = \frac{1}{2N} \|D\theta - \mathbf{y}\|^2,$$

where $\mathbf{y} = [y_1 \quad y_2 \quad \cdots \quad y_N]^\top$ is the label vector. We can reorder the terms so that

$$F(\theta) = \frac{1}{2N}(D\theta - y)^\top(D\theta - y) = \frac{1}{2N}\left(\theta^\top D^\top D\theta - 2\theta^\top X^\top \mathbf{y}\right) + \text{const.}$$

$$= \frac{1}{2}(\theta - \widehat{\theta})^\top \frac{D^\top D}{N}(\theta - \widehat{\theta}) + \text{const.}$$

where $\widehat{\theta} = (D^\top D)^{-1} D^\top \mathbf{y}$. Note that this is precisely the ordinary least squares solution.

Since the loss function is quadratic, we can explicitly write the Gibbs distribution (which is the unrestricted solution to the Bayesian Learning Problem with $F$) as the Gaussian distribution

$$q^*(\theta) \propto e^{-\beta F(\theta)} \propto e^{-\frac{1}{2}(\theta - \widehat{\theta})^\top \frac{D^\top D}{N/\beta}(\theta - \widehat{\theta})} \quad \text{thus } q^*(\theta) = \mathcal{N}\left(\widehat{\theta}, \left(\frac{D^\top D}{N/\beta}\right)^{-1}\right).$$

Here, the variable $\beta$ is the inverse temperature defined as $\beta = 1/\tau$.

Next, we construct $G$, a loss function on $\mathcal{X} \times \mathcal{Y}$. By fixing the label, we may also consider $G$ as a loss function only on $\mathcal{X}$, from which we derive a distribution over $\mathcal{X}$. To avoid overusing $\mathbf{x}$ and $y$, we denote elements of the labeled dataset as $(\mathbf{z}, w) \in \mathcal{X} \times \mathcal{Y}$ with $\mathbf{z} = \begin{bmatrix} \mathbf{f} \\ 1 \end{bmatrix}$. Using the first and second moments of Gaussians, we calculate

$$G(\mathbf{z}, w) = \int_\Theta |\mathbf{z}^\top \theta - w|^2 q^*(\theta) \mathrm{d}\,\theta$$

$$= \mathbf{z}^\top \mathbb{E}_{q^*}[\theta \theta^\top] \mathbf{z} - 2w\mathbf{z}^\top \mathbb{E}_{q^*}[\theta] + \text{ const}$$

$$= \mathbf{z}^\top \left(\widehat{\theta}\widehat{\theta}^\top + \left(\frac{D^\top D}{N\tau}\right)^{-1}\right) \mathbf{z} - 2w\mathbf{z}^\top \widehat{\theta} + \text{ const.}$$

which is again a quadratic function in $\mathbf{z}$. Let us now write this quadratic in terms of $\mathbf{f}$. We write $\widehat{\theta} = \begin{bmatrix} \widehat{\boldsymbol{\xi}} \\ \widehat{b} \end{bmatrix}$.

First, a quick calculation gives the block diagonal form

$$
\left(\frac{D^\top D}{N\tau}\right)^{-1} = \tau \left[\begin{array}{c|c} \frac{X^\top X}{N} & \overline{\mathbf{x}} \\ \hline \overline{\mathbf{x}}^\top & 1 \end{array}\right]^{-1}
$$

$$
= \tau \left[\begin{array}{c|c} A^{-1} & -A^{-1}\overline{\mathbf{x}} \\ \hline -\overline{\mathbf{x}}^\top A^{-1} & * \end{array}\right],
$$

where $A = \frac{X^\top X}{N} - \overline{\mathbf{x}}\overline{\mathbf{x}}^\top$ is the Schur complement and $\overline{\mathbf{x}} = \frac{1}{N}\sum_{i=1}^N \mathbf{x}_i$ is the mean data vector.

We can write $G$ as a quadratic function of $\mathbf{f}$ (fixing $w$) as

$$
G_w(\mathbf{f}) = \mathbf{f}^\top \left(\tau A^{-1} + \widehat{\boldsymbol{\xi}}\widehat{\boldsymbol{\xi}}^\top\right)\mathbf{f} - 2\mathbf{f}^\top\left(\tau A^{-1}\overline{\mathbf{x}} - \widehat{\boldsymbol{\xi}}\hat{b} + w\hat{b}\right) + \text{const.}
$$

$$
= (\mathbf{f} - \widehat{\mathbf{f}})\left(\tau A^{-1} + \widehat{\boldsymbol{\xi}}\widehat{\boldsymbol{\xi}}^\top\right)(\mathbf{f} - \widehat{\mathbf{f}}) + \text{const.}
$$

Here, $\widehat{\mathbf{f}}$ is calculated as

$$
\widehat{\mathbf{f}} = \left(\tau A^{-1} + \widehat{\boldsymbol{\xi}}\widehat{\boldsymbol{\xi}}^\top\right)^{-1}\left(\tau A^{-1}\overline{\mathbf{x}} + \widehat{\boldsymbol{\xi}}(w - \hat{b})\right)
$$

$$
= \left(A_\tau - \frac{A_\tau\widehat{\boldsymbol{\xi}}\widehat{\boldsymbol{\xi}}^\top A_\tau}{1 + \widehat{\boldsymbol{\xi}}^\top A_\tau\widehat{\boldsymbol{\xi}}}\right)\left(A_\tau^{-1}\overline{\mathbf{x}} + \widehat{\boldsymbol{\xi}}^\top(w - \hat{b})\right),
$$

where $A_\tau = \frac{1}{\tau}A$ and the Sherman-Morrison formula is used for inverting the matrix.

Now expanding the product, we obtain

$$
\widehat{\mathbf{f}} = \overline{\mathbf{x}} + A_\tau\widehat{\boldsymbol{\xi}}(w - \hat{b}) - \frac{A_\tau\widehat{\boldsymbol{\xi}}\widehat{\boldsymbol{\xi}}^\top\overline{\mathbf{x}}}{1 + \widehat{\boldsymbol{\xi}}^\top A_\tau\widehat{\boldsymbol{\xi}}} - A_\tau\widehat{\boldsymbol{\xi}}\frac{\widehat{\boldsymbol{\xi}}^\top A_\tau\widehat{\boldsymbol{\xi}}}{1 + \widehat{\boldsymbol{\xi}}^\top A_\tau\widehat{\boldsymbol{\xi}}}(w - \hat{b}).
$$

Note that if we denote the predictions of the linear model as $\mathbf{x}_i^\top\widehat{\boldsymbol{\xi}} + \hat{b} = \hat{y}_i$, we can rewrite the above formula as follows:

$$
\widehat{\mathbf{f}} = \overline{\mathbf{x}} + A_\tau\widehat{\boldsymbol{\xi}}(w - \hat{b})\frac{1}{1 + \hat{\boldsymbol{\xi}}^\top A_\tau\hat{\boldsymbol{\xi}}} - A_\tau\hat{\boldsymbol{\xi}}\frac{\hat{\boldsymbol{\xi}}^\top\overline{\mathbf{x}}}{1 + \hat{\boldsymbol{\xi}}^\top A_\tau\hat{\boldsymbol{\xi}}}
$$

$$
= \overline{\mathbf{x}} + A_\tau\hat{\boldsymbol{\xi}}\frac{(w - \hat{b})}{1 + \hat{\boldsymbol{\xi}}^\top A_\tau\hat{\boldsymbol{\xi}}} - A_\tau\hat{\boldsymbol{\xi}}\frac{(\widehat{\overline{\mathbf{y}}} - \hat{b})}{1 + \hat{\boldsymbol{\xi}}^\top A_\tau\hat{\boldsymbol{\xi}}}
$$

$$
= \overline{\mathbf{x}} + A_\tau\hat{\boldsymbol{\xi}}\frac{w - \widehat{\overline{\mathbf{y}}}}{1 + \hat{\boldsymbol{\xi}}^\top A_\tau\hat{\boldsymbol{\xi}}}.
$$

Here, we denoted the prediction of the average data by $\widehat{\overline{\mathbf{y}}} = \hat{\boldsymbol{\xi}}^\top\overline{\mathbf{x}} = \frac{1}{N}\sum_{i=1}^N \hat{y}_i$.

Finally, let's rewrite $A_\tau\widehat{\boldsymbol{\xi}}$ and $\widehat{\boldsymbol{\xi}}^\top A_\tau\widehat{\boldsymbol{\xi}}$ in terms of interpretable statistical quantities. Recall that $A_\tau = \frac{1}{\tau}\left(\frac{X^\top X}{N} - \overline{\mathbf{x}}\overline{\mathbf{x}}^\top\right)$. Using this, we compute

$$
A_\tau\widehat{\boldsymbol{\xi}} = \frac{1}{\tau}\left(\frac{1}{N}\sum_{i=1}^N \mathbf{x}_i(\underbrace{\mathbf{x}_i^\top\widehat{\boldsymbol{\xi}}}_{=\hat{y}_i - \hat{b}}) - \overline{\mathbf{x}}\underbrace{\overline{\mathbf{x}}^\top\hat{\boldsymbol{\xi}}}_{=\frac{1}{N}\sum_{i=1}^N \hat{y}_i - b}\right)
$$

$$
= \frac{1}{N\tau}\sum_{i=1}^N (\mathbf{x}_i - \overline{\mathbf{x}})(\hat{y}_i - \hat{b})
$$

$$
= \frac{1}{N\tau}\sum_{i=1}^N (\mathbf{x}_i - \overline{\mathbf{x}})(\hat{y}_i - \widehat{\overline{\mathbf{y}}})
$$

$$
= \frac{1}{\tau}\text{Cov}(X, \widehat{\mathbf{y}}).
$$

In the final expression, the term $\mathrm{Cov}(X, \widehat{\mathbf{y}})$ corresponds directly to the previous line. This covariance is a vector that averages data deviations, weighted by prediction deviations. In the line before last, we replaced $\widehat{b}$ with any constant since it is independent of $i$, and the first factor sums to the zero vector. Additionally, we leveraged a key property of linear models: the average of the predictions is the same as the prediction of the average.

A similar calculation yields,

$$\hat{\boldsymbol{\xi}}^{\top} A_\tau \hat{\boldsymbol{\xi}} = \frac{1}{\tau} \left( \frac{1}{N} \sum_{i=1}^{N} \hat{y}_i^2 - \left( \frac{1}{N} \sum_{i=1}^{N} \hat{y}_i \right)^2 \right) = \frac{1}{\tau} \mathrm{Var}(\widehat{\mathbf{y}}).$$

Therefore, we obtain an explicit quadratic formulation of the data loss function $G$ in terms of $\mathbf{f}$ at a fixed $w$. This means that the data distribution $p^*(\mathbf{x})$, which solves the unrestricted Bayesian Learning Problem, follows a Gaussian distribution given as

$$p^*(\mathbf{f}) \propto e^{-G_w(\mathbf{f})} \propto \mathcal{N}(\widehat{\mathbf{f}}, \Sigma),$$

where

$$\widehat{\mathbf{f}} = \overline{\mathbf{x}} + \frac{\mathrm{Cov}(X, \widehat{\mathbf{y}})}{\tau + \mathrm{Var}(\widehat{\mathbf{y}})} \left( w - \frac{1}{N} \sum_{i=1}^{N} \widehat{y}_i \right),$$

and

$$\Sigma^{-1} = \left( \tau \left( \frac{X^\top X}{N} - \overline{\mathbf{x}}\overline{\mathbf{x}}^\top \right)^{-1} + \widehat{\boldsymbol{\xi}}\widehat{\boldsymbol{\xi}}^\top \right).$$

The interpretation of the mean $\widehat{\mathbf{f}}$ is as follows: if you want to sample from a data distribution that will produce a given $\omega$, then you should not sample around $\overline{\mathbf{x}}$ (which would be the case without output restrictions). Instead, you shift $\overline{\mathbf{x}}$ in proportion to the difference between $\omega$ and the mean of the training label predictions, following the direction of the covariance between the training data and predicted labels.

# B    COMPUTATIONAL SETUP

In this section, we supplement our computational study by presenting the datasets and detailing the neural network architectures.

## B.1    DATASETS USED IN THE EXPERIMENTS

Our experiments are conducted using four numerical datasets and one visual dataset from the literature. The details of the datasets are provided below.

**Adult.** The Adult dataset (CC BY 4.0 license), derived from the 1994 Census database, comprises 48,842 observations with 14 features, including both continuous and categorical variables (Becker & Kohavi, 1996). The primary objective is to classify individuals based on whether their annual income exceeds $50,000 USD. Data preprocessing steps are applied to address missing values and handle categorical features. We applied one-hot encoding to transform the categorical features into a numerical format suitable for our framework.

**FICO.** The FICO (HELOC) dataset (CC0: Public Domain) consists of home equity line of credit applications submitted by homeowners (FICO, 2018). It includes 10,459 records with 23 features, comprising both numerical and ordinal variables. The primary objective is to classify applications based on their risk performance, identifying whether an applicant is likely to meet payment obligations or become delinquent. Data preprocessing steps are applied to address missing values.

**Housing.** The Housing dataset (CC0: Public Domain), sourced from Kaggle, includes information on various house attributes such as lot size, number of rooms, and number of stories (Kaggle, 2021). The dataset contains 535 records and 12 features, comprising both numerical and ordinal variables. The primary objective is to predict housing prices based on these features.

**MNIST.** The MNIST dataset (Data files © Original Authors) is a widely used benchmark in computer vision, consisting of 70,000 grayscale images of handwritten digits (0–9), each represented as a 28×28 pixel matrix (LeCun et al., 2010). The dataset is divided into 60,000 training samples and 10,000 test samples. The primary objective is to classify images based on the digit they represent. We normalized each of the images to be arrays of shape $(28, 28, 1)$ with FP32 values in the interval $[0, 1]$.

**Wine Recognition Dataset.** The Wine Recognition dataset contains the results of chemical and physical analyses of wines produced in three regions of Italy (Aeberhard & Forina, 1992). Each record is described by 13 continuous features obtained through chemical analysis, including alcohol content, malic acid, magnesium, color intensity, and hue. The primary objective is to classify wines according to their region of origin.

### B.2 EXPERIMENTAL SETUP

We ran all the experiments on a personal Apple M1 Pro with 32 GB RAM. In our experiments compute time was not a significant factor, all experiments were completed within minutes.

For the parameter-sensitive and prediction-risky experiments on the FICO dataset, we trained an MLP with ReLU activation functions and layer widths of $128 - 32 - 8 - 2$. Dropout with a rate of 0.2 was applied after each activation layer to prevent overfitting. The model was trained using a batch size of 128 for $10,000$ steps.

For the image experiments, we used an MLP with layer widths of 1024-128-10, where each layer included a ReLU activation, followed by a dropout layer with a rate of 0.2. The CNN architecture consisted of two convolutional blocks with feature sizes $32 - 64$. Each block followed the structure: Conv $\rightarrow$ ReLU $\rightarrow$ Conv $\rightarrow$ ReLU $\rightarrow$ max_pool $\rightarrow$ Dropout, where the convolutional kernels had a size of $3 \times 3$, the max pooling window was $2 \times 2$, and the dropout rate was 0.2.

Both the CNN and MLP models were trained for 10,000 update steps using a batch size of 128 and the Adam optimizer. The learning rate followed an exponential decay schedule, starting with a maximum learning rate of 0.1, decaying by a rate of 0.9 every 100 steps.

## C LANGEVIN DYNAMICS SOLVING THE BAYESIAN LEARNING PROBLEM

In solving equation 3, we sample data points by solving a Bayesian optimization problem over distributions on the data space:

$$p^* = \arg\min_{p \in \mathcal{Q}} \mathbb{E}[G] - \tau \mathcal{H}(p),$$

where $\mathcal{H}(p)$ denotes the entropy and $\tau > 0$ is a temperature parameter.

The intuition behind this objective is that instead of minimizing a function $G$ we find a distribution over the space $X$ which balances having its distributional mass focused on points where $G$ is low and also a spread widely. This objective is also called the Evidence Lower BOund (ELBO) and interpreting the loss as negative log-likelihood the solution can be interpreted as Bayes's formula updating beliefs upon observing new data, as shown by Zellner Zellner (1988).

### C.1 METROPOLIS HASTINGS LANGEVIN ALGORITHM (MALA)

On unconstrained data space, *i.e.*, when $\mathcal{Q} = \mathcal{P}(\mathbb{R}^d)$, one approach of sampling from $p^*$ is to simulate the Langevin dynamics, which is a Stochastic Differential Equation (SDE) given by

$$\frac{\mathrm{d}X}{\mathrm{d}t} = -\frac{1}{\tau} \nabla G(X) + \sqrt{2} \frac{\mathrm{d}W}{\mathrm{d}t},$$

where $W$ is a Wiener process (*i.e.*, standard Brownian motion), the $G$ is called a drift term. The limiting distribution of this SDE converges to $p_\infty(X) \propto e^{-\frac{1}{\tau}G(x)}$.

For concrete computation, we work with the standard Euler-Maruyama discretization

$$\tilde{\mathbf{x}}_{t+1} = \mathbf{x}_t - \eta \nabla G(\mathbf{x}_t) + \sqrt{2\eta\tau}\varepsilon_t,$$

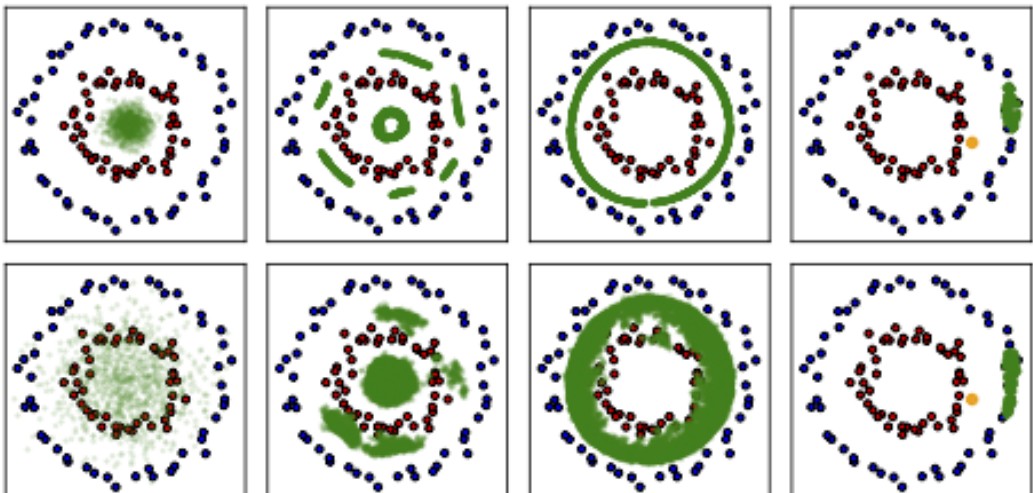

Figure 8: The probing functions in Figure 2, where in the first there columns the data generation in the second line trained with different temperatures. The last column shows the effect of a $\ell_2$ regularizing term (with anchor point shown in orange) with a smaller strength.

where $\eta > 0$ is the step size, and $\varepsilon_t \sim \mathcal{N}(\mathbf{0}, \mathbf{I}_d)$ is sampled independently from the unit normal distribution at each time step.

To improve convergence and ensure the correctness of the sampling distribution, we apply a Metropolis–Hastings acceptance step. Specifically, the proposed update $\mathbf{x}_{t+1}$ is accepted as $\mathbf{x}_{t+1} = \tilde{\mathbf{x}}_{t+1}$ with probability

$$\alpha := \min\left\{1, \frac{\pi(\tilde{\mathbf{x}}_{t+1})q(\mathbf{x}_t|\tilde{\mathbf{x}}_{t+1})}{\pi(\mathbf{x}_t)q(\tilde{\mathbf{x}}_{t+1}|\mathbf{x}_t)}\right\}$$

or otherwise rejected, in which case $\mathbf{x}_{t+1} = \mathbf{x}_t$. Here, $\pi \propto e^{-\frac{1}{\tau}G}$ and $q(\mathbf{x}'|\mathbf{x})$ is the transition probability of stepping from $\mathbf{x}$ to $\mathbf{x}'$ given by $q(\mathbf{x}'|\mathbf{x}) = \exp(-\frac{1}{4\tau}\|\mathbf{x}' - \mathbf{x} + \tau\nabla G(\mathbf{x})\|^2)$. An implementation of this sampling procedure is provided in the submitted code repository, specifically in the file `langevin.py` using JAX/PyTorch and Numpy.

The temperature hyperparameter $\tau$ determines the balance between exploration of the data space versus minimizing $G$. In the figure below we show the effect of changing this parameter (as well as the effect of changing the tunable parameter of the regularizing term).

## C.2 Smoothing

The acceptance/rejection is the critical step to ensure convergence of the trajectory to the limiting Gibbs-Boltzmann distribution $\propto e^{-\frac{1}{\tau}G}$. The drift term $\nabla G(\mathbf{x})$ speeds up the convergence by ensuring that more of the proposals $\tilde{\mathbf{x}}$ will be accepted since this becomes (noisy) gradient descent.

In case the function $G$ is locally flat, such as those functions created from tree-based models (Decision Tree, Random Forest, XGBoost) then the gradient term is always $\mathbf{0}$, reducing the proposals to simple random walk. This is called the Metropolis Hastings (MH) algorithm and it also has the same limiting distribution, and the transition probabilities in the acceptance ratio cancel since $q(\mathbf{x}|\mathbf{x}') = q(\mathbf{x}'|\mathbf{x})$.

However in this case at any step the proposals are random and the point $\mathbf{x}$ does not see if it is near a decision boundary or not. For this purpose we propose using the gradients of the smoothed function $G_s(\mathbf{m}) = \int G(\mathbf{x})\mathcal{N}(\mathbf{m}, \sigma)(\mathbf{x})\mathrm{d}\mathbf{x}$ for the proposals. With a larger $\sigma$, the point $\mathbf{x}$ "sees" a wider horizon for its proposals. Indeed the gradient of $G_s$ is

$$\nabla G_s(\mathbf{x}) = \int G(\mathbf{x} + \sigma\varepsilon)\varepsilon\mathcal{N}(\mathbf{0}, I)(\varepsilon)\mathrm{d}\varepsilon \approx \frac{1}{J}\sum_{j=1}^{J} G(\mathbf{x} + \sigma\varepsilon_j)\varepsilon_j, \qquad \text{with } \varepsilon_j \sim \mathcal{N}(\mathbf{0}, I).$$

In our implementation, we applied proposals with $\nabla G_s$ and acceptance/rejection with $G$ In the code, we also assumed $\nabla G_s(\mathbf{x}) = \nabla G_s(\mathbf{x}')$ for computational simplicity in calculating the transition probabilities $q(\mathbf{x}'|\mathbf{x})$. This is approximately true when $G$ is a step function and the step size $\eta$ is small.

### C.3 OTHER METHODS AND VARIANTS

Beyond the Langevin dynamics approach, several alternative methods exist for sampling data points. For instance, one may use the Picard iteration method or various splitting schemes (see (Geiser, 2020)) to discretize and solve the above SDE. Another approach is to restrict the distribution family $\mathcal{Q}$ to be a tractable statistical manifold and apply Variational Inference techniques (Ganguly & Earp, 2021).

It is also worth noting that when sampling over a data manifold, if there exists a mapping $\varphi : \mathbb{R}^d \to \mathcal{X}$ and $G : \mathcal{X} \to \mathbb{R}$, then the corresponding Gibbs distribution on $\mathbb{R}^d$ (with respect to the Lebesgue measure) is given by $e^{-\frac{1}{\tau}H}$, where $H = G \circ \varphi : \mathbb{R}^d \to \mathbb{R}$. Samples drawn from this distribution can be pushed forward via $\varphi$ to obtain samples in $\mathcal{X}$. These pushed-forward samples follow a distribution that can be interpreted as a Gibbs distribution over $\mathcal{X}$ with respect to the base measure $\nu$, which is the pushforward of the Lebesgue measure under $\varphi$.

## D ADDITIONAL NUMERICAL RESULTS

This section presents additional results that complement the findings discussed in Section 4. These results provide further insights into the generated data distributions, feature variations, and model behavior under different probing scenarios.

**Model-contrasting samples.** This subsection extends our analysis of model-contrasting samples by applying the framework to a different tabular dataset. In this experiment, we examine prediction divergence between support vector regression (SVR) and linear regression (LR) models. We use the Housing dataset, where the primary objective is to predict house prices based on various structural and amenity-related features. We split the dataset into training-test sets and train both models on the same training data. To generate data samples where the two models diverge in their predictions, we formulate the cost function given in (5) as $\ell_G(y_1, 1 - y_2) = \exp(-(y_1 - y_2)^2)$. Using our framework, we generate data samples to identify the regions of the input space where the models exhibit significant disagreement, likely due to their differing assumptions about feature interactions and predictive mechanisms.

Figure 9 presents a scatter plot comparing the predictions of the SVR and LR models. The blue points represent the predictions of the models in the test data, demonstrating that the two models generally produce highly similar outputs, with minimal differences observed. The green points, on the other hand, represent generated samples, highlighting instances where the models exhibit contrasting predictions. The zoomed-in inset further emphasizes these discrepant predictions, demonstrating that our framework effectively identifies and generates data points that maximize the divergence between the two models.

Figure 10 compares the feature distributions between the synthetic dataset generated by our framework and the test data. The box plots represent the range of values for each feature, with blue corresponding to the test data and green representing the generated samples. The Housing dataset (Kaggle, 2021) used in this experiment contains real estate information such as lot size, number of bathrooms, number of stories, and heating/air conditioning types, aiming to predict house prices. This figure provides a clear visualization of how the generated data differs from the test data in terms of feature distributions. For instance, as the number of bathrooms and stories increases, the model predictions diverge. Additionally, hot water heating and air conditioning exhibit a distinct concentration in the synthetic data, with most generated samples clustering around higher values compared to the test data. This suggests that these features play a prominent role in distinguishing instances where the models behave differently. Overall, this figure offers insights into how the generated samples differ from the original dataset, highlighting key feature distributions that drive divergence in model predictions and providing a deeper understanding of how our framework probes model behavior.

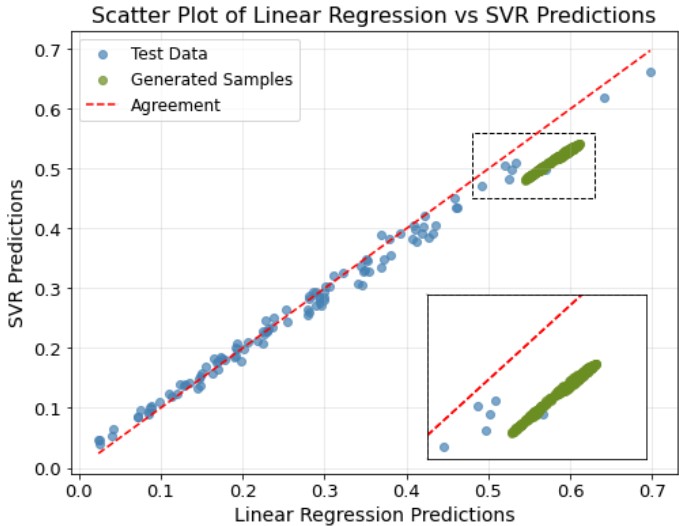

Figure 9: Comparison of SVR and LR predictions on test and generated data.

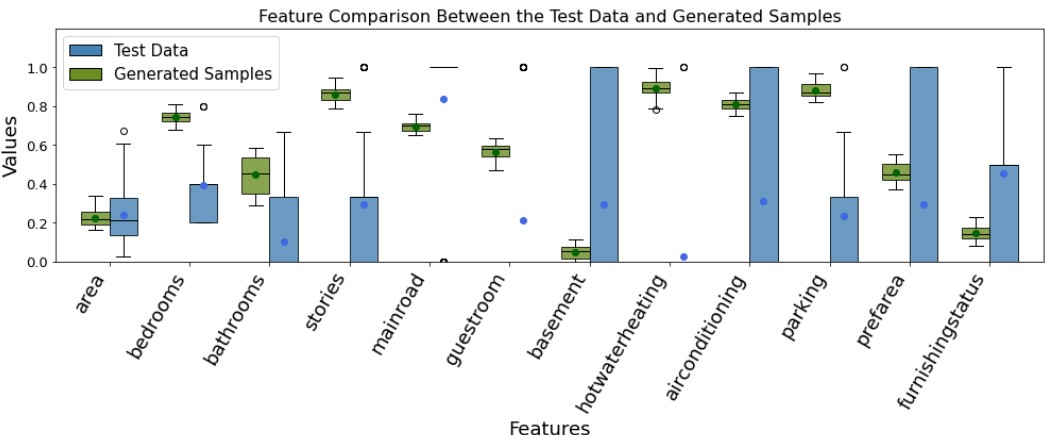

Figure 10: Features in test data and generated samples that produce different predictions for SVR and LR.

**Prediction-risky samples.** This experiment explores data samples near the model's decision boundary, where predictions are inherently uncertain. To guide this analysis, we pose the question:

*Which data samples are predicted to be risky due to being close to a specific anchor value?*

In this experiment, we train a neural network (MLP) to classify customers in the FICO dataset as either "Good" or "Bad" credit risks. Prediction-risky samples are those with model outputs near the anchor value of 0.5. Using our framework, we generate 500 such samples to examine the characteristics of borderline classification cases. The average predicted probability of the "Bad" credit class among these samples is 0.525, with a standard deviation of 0.017.

The density plots in Figure 11 compare the distributions of two representative features in the original data and the generated samples. The feature `NumTrades60Ever2DerogPubRec` represents the number of past credit trades with payments delayed by at least 60 days, serving as a key indicator of delinquency. As shown in the figure, the distribution of risky samples follows the original data closely in the lower range but exhibits a stronger peak around zero. This suggests that the model considers individuals with few or no past delinquencies as borderline cases, likely due to the absence of strong negative or positive indicators, making classification more uncertain. The feature

`MSinceOldestTradeOpen` indicates the number of months since a customer's first credit line was opened, capturing the length of their credit history. As shown in Figure 11, the distribution of risky samples is sharply concentrated around 400 months (~33 years), whereas the original data spans a broader range. This suggests that the model associates long credit histories with greater uncertainty. The pronounced peak reflects how the model treats long-established credit profiles as ambiguous when making predictions.

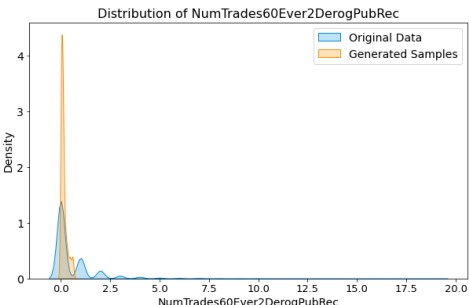 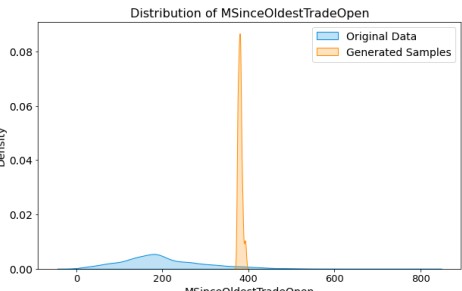

Figure 11: Feature distributions of the original data and generated prediction-risky samples.

To further investigate data samples near the decision boundary, we present the distributions of all features in the original dataset and the generated prediction-risky samples in Figure 12. These density plots provide a comprehensive view of the differences between the generated samples and the original data across multiple features. By analyzing these distributions, we can observe how the model identifies borderline cases based on different financial attributes. Across multiple features, the generated prediction-risky samples exhibit a much narrower distribution compared to the original data. This suggests that the model focuses on a specific subset of feature values when identifying borderline cases.

Table 1: Characteristics of the wine recognition dataset and the generated data

| Characteristic | Class 0 | Class 1 | Class 2 | Generated |
|---|---|---|---|---|
| **Samples** | 59 | 71 | 48 | 50 |
| Alcohol | $13.74_{\pm 0.63}$ | $12.28_{\pm 0.53}$ | $13.15_{\pm 0.52}$ | $12.75_{\pm 0.61}$ |
| Malic acid | $2.01_{\pm 0.68}$ | $1.93_{\pm 1.00}$ | $3.33_{\pm 1.08}$ | $1.59_{\pm 0.46}$ |
| Ash | $2.46_{\pm 0.22}$ | $2.45_{\pm 0.31}$ | $2.44_{\pm 0.18}$ | $1.83_{\pm 0.25}$ |
| Alcalinity of ash | $17.04_{\pm 2.52}$ | $20.24_{\pm 3.33}$ | $21.42_{\pm 2.23}$ | $19.37_{\pm 4.95}$ |
| Magnesium | $106.34_{\pm 10.41}$ | $94.55_{\pm 16.63}$ | $99.31_{\pm 10.78}$ | $98.9_{\pm 28.91}$ |
| Total phenols | $2.84_{\pm 0.34}$ | $2.26_{\pm 0.54}$ | $1.68_{\pm 0.35}$ | $1.69_{\pm 0.43}$ |
| Flavanoids | $2.98_{\pm 0.39}$ | $2.08_{\pm 0.7}$ | $0.78_{\pm 0.29}$ | $2.11_{\pm 0.82}$ |
| Nonflavanoid phenols | $0.29_{\pm 0.07}$ | $0.36_{\pm 0.12}$ | $0.45_{\pm 0.12}$ | $0.3_{\pm 0.08}$ |
| Proanthocyanins | $1.90_{\pm 0.4}$ | $1.63_{\pm 0.6}$ | $1.15_{\pm 0.4}$ | $2.46_{\pm 0.41}$ |
| Color intensity | $5.53_{\pm 1.23}$ | $3.09_{\pm 0.92}$ | $7.4_{\pm 2.3}$ | $2.34_{\pm 0.6}$ |
| Hue | $1.06_{\pm 0.16}$ | $1.06_{\pm 0.2}$ | $0.68_{\pm 0.11}$ | $1.27_{\pm 0.14}$ |
| OD280/OD315 | $3.16_{\pm 0.35}$ | $2.78_{\pm 0.49}$ | $1.68_{\pm 0.27}$ | $1.69_{\pm 0.39}$ |
| Proline | $1115.71_{\pm 221.64}$ | $519.5_{\pm 156.1}$ | $629.9_{\pm 113.9}$ | $1200.8_{\pm 233.3}$ |

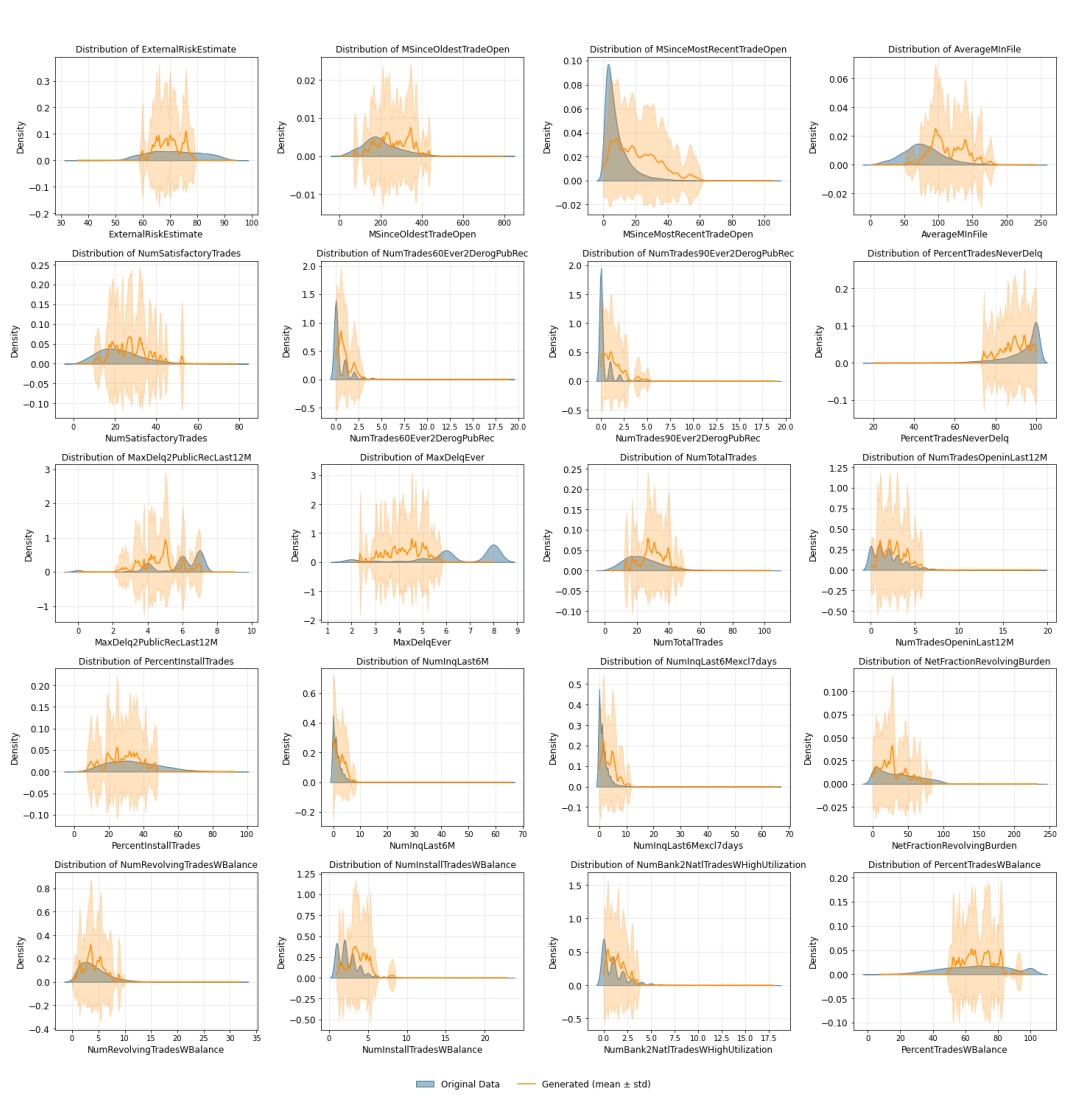

Figure 12: Feature distributions in the original data and generated prediction-risky samples.

**Parameter-sensitive samples.** To complement the findings presented in Section 4, we provide the full set of feature distributions comparing parameter-sensitive samples and prediction-risky samples in Figure 13. These density plots illustrate how the two types of generated samples differ. By analyzing these distributions, we observe that while some features exhibit similar trends across both sample types, others show notable divergences. Features with broader distributions in parameter-sensitive samples indicate that model perturbations impact a wider range of instances.

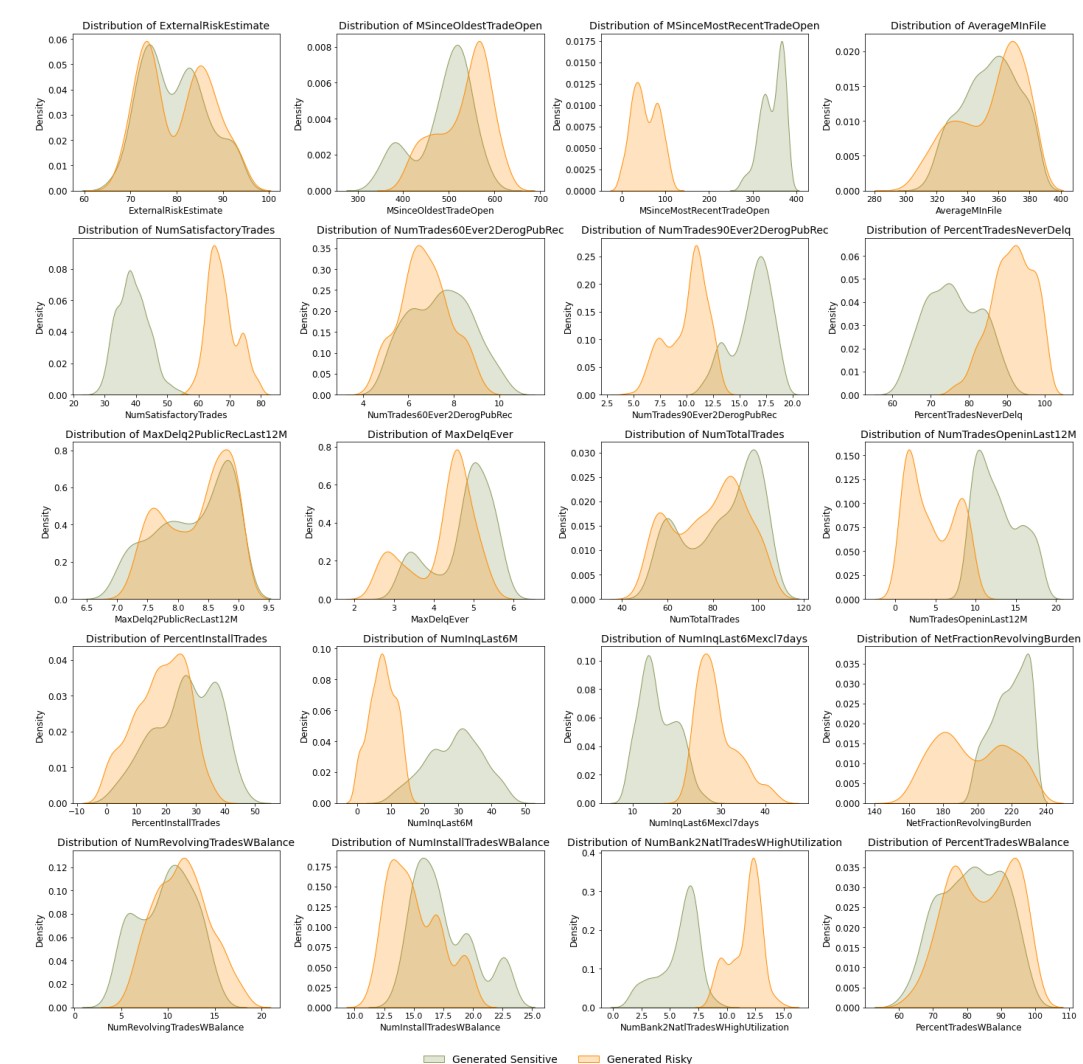

Figure 13: Feature distributions in generated parameter-sensitive and prediction-risky samples.

**Fixed-label samples.** We now analyze a different factual instance from the original data to further investigate the model's behavior. The factual instance considered represents a Latin American white male who is predicted to earn more than $50K. To explore the conditions under which the model would classify this individual as earning less than $50K, we generate a set of counterfactual samples. Figure 14 presents the distribution of these generated counterfactual samples, highlighting the key feature variations that lead to a different classification outcome. In the generated counterfactual samples, while no categorical changes are observed, the numerical features age, educational attainment, and working hours exhibit lower values compared to the factual instance, implying that a reduction in these features leads to a shift in classification.

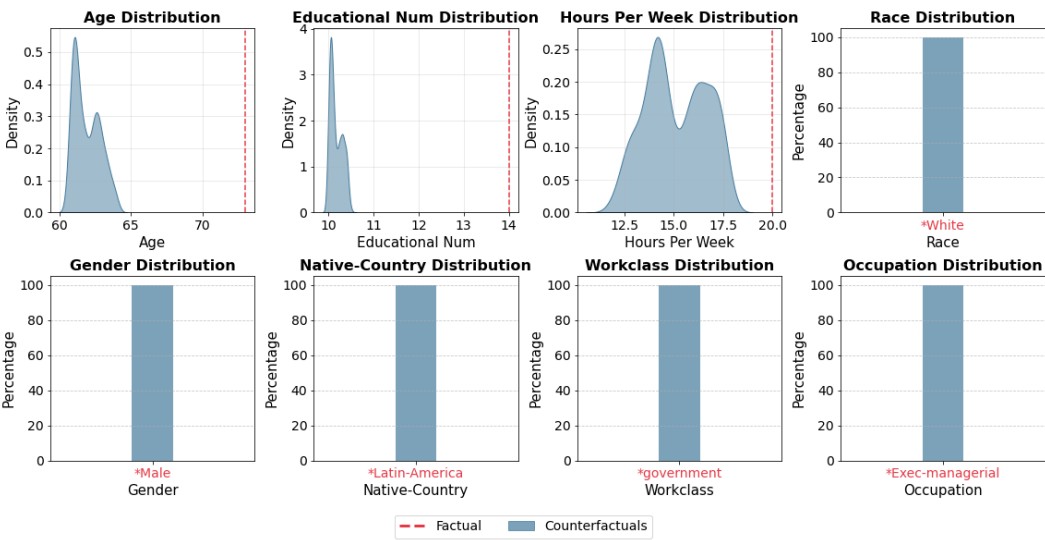

Figure 14: Feature distributions of generated counterfactual samples (blue shaded) with factual instance highlighted (red markers).

Additionally, we compare the counterfactual distribution produced by our probing scenario with counterfactual examples generated by DiCE for a representative Adult dataset instance. In this experiment, the factual instance represents a North-American White Female, predicted to earn less than $50K. Figure 15 shows the marginal feature distributions for DiCE counterfactuals, our generated counterfactual samples, and the original training data, together with the factual instance. As expected, DiCE produces a finite set of discrete counterfactuals with relatively high variance across many features, and often explores extreme or low-density regions of the input space. In contrast, our method generates a full stationary distribution over counterfactual inputs, producing tightly clustered samples that remain stable across independent runs. The resulting distribution concentrates on plausible, high-density regions of the dataset and alters only the features necessary for flipping the model's decision. We note that all generated samples strictly satisfy the decision-flipping condition. This highlights the complementary nature of the two approaches. DiCE provides diverse point-wise counterfactuals, while our probing scenario characterizes the underlying counterfactual landscape by modeling the distribution of model-aligned inputs.

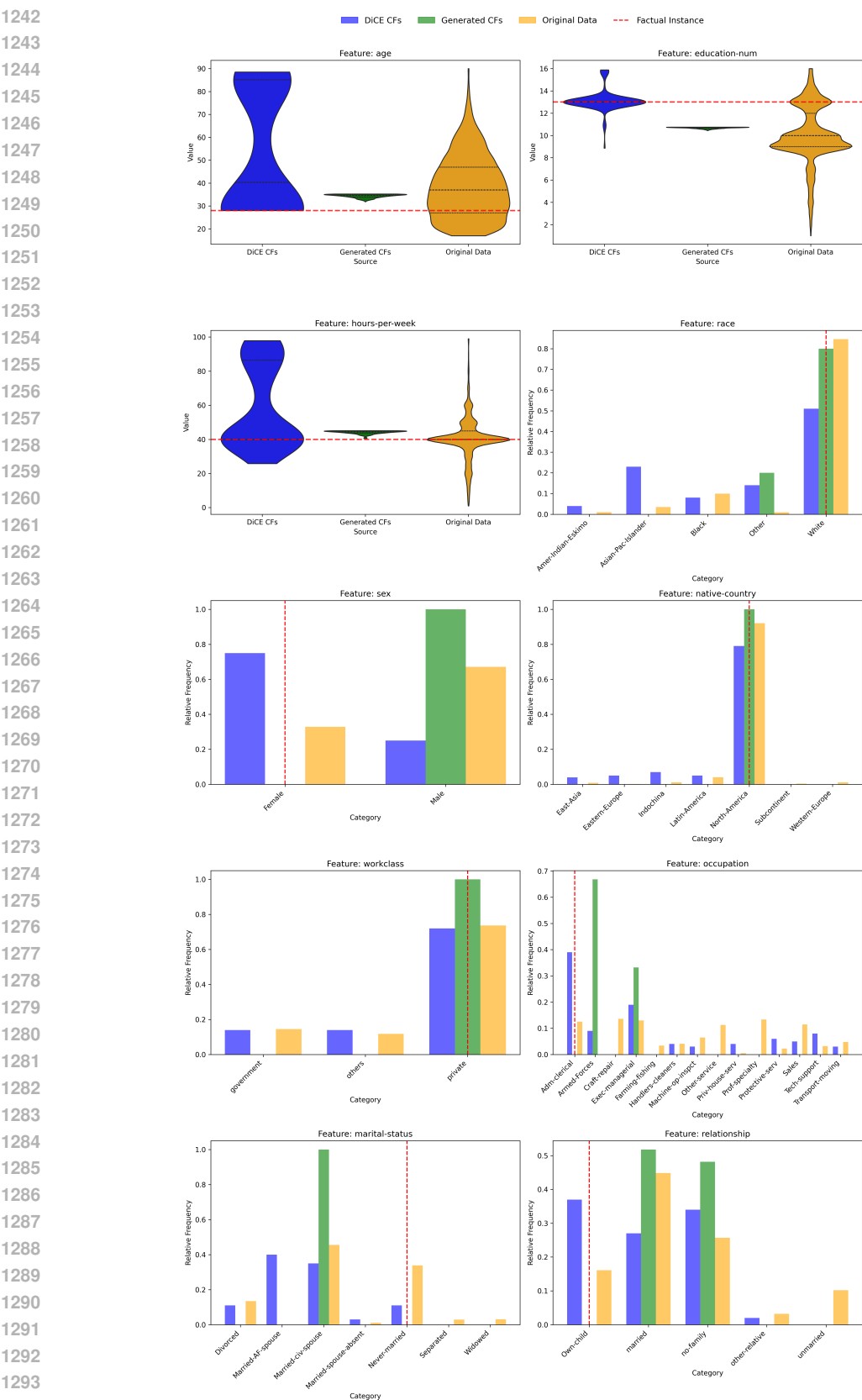

Figure 15: Comparison between our generated counterfactual distribution and DiCE counterfactual examples for the same factual instance.

## E   THE USE OF VAEs

A notable example of using pushforwards to obtain points on the data manifold comes from image datasets. We employ a VAE architecture with two convolutional layers each for the encoder and decoder submodules. Features in the convolutional layers are 32 and 64 with kernel sizes of (3,3) and a stride of (2,2). During training, the reconstruction loss is computed using bitwise entropy.

Figure 16 shows how this setup works for constructing loss functions $G$ on the latent space. One may use a combination of models, each precomposed with the decoder of the trained VAE. The resulting distribution on the latent space, after pushforwarding (*i.e.*, passing the samples through the decoder), corresponds to a distribution on the data that is closer to the original data distribution.

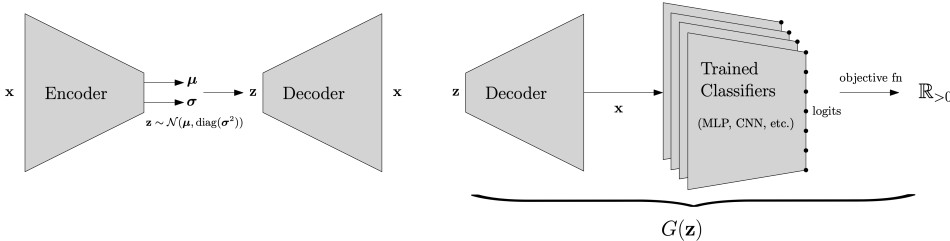

Figure 16: By precomposing with the decoder submodule of a trained neural network, we can define $G$ functions on the lower-dimensional latent space, while still leveraging networks designed for higher-dimensional image inputs.

## F   HIGH DIMENSIONAL IMAGE GENERATION

To generate the images in Figure 7, using a VAE alone is insufficient for producing images of size $256 \times 256$. While a VAE ensures local consistency of colors in small patches, it does not guarantee global coherence. To address this, we successively increase the resolution of generated samples, making use of the TAESD autoencoder, which supports multiple input sizes.

The latent space of the TAESD has shape $16 \times \frac{n}{8} \times \frac{m}{8}$, where $n \times m$ is the image size. At each resolution (e.g., $64 \times 64$), we run MALA according to a probing function $G$. This function $G$ simultaneously (1) reduces the cross entropy loss of the logits produced by decoder followed by the ResNet50 classifier, (2) minimizes the reconstruction loss of the high dimensional image corresponding to the latent vector, and (3) enforces closeness to the previously generated latent vector corresponding to the image at the lower resolution (e.g., $32 \times 32$). This procedure of successively increasing the resolution of the image, and starting from the resized version of the previous MALA run, maintains global consistency of the generated images whilst also satisfying the probing function requirements. Implementation details are provided in the `probe_resnet.py` script in our repository.

In Figure 7, we show samples generated with $G$ function aiming to minimize the cross entropy loss between the ResNet50 logits and the label corresponding to goldfish:1, and snail:113. Each of the images in the figure has resolution $256 \times 256$.

By qualitatively observing these figures, we were able to form a hypothesis on how the pretrained ResNet50 (using the weights `ResNet50_Weights.IMAGENET1K_V2` from `torchvision.models`) detects the class label 1 corresponding to goldfish. The presence of the color orange in every image and also a black dot inside the orange corresponding to the eye of the fish was present in most of the images. In order to test the color hypothesis, we modify the 50 goldfish validation images from ImageNet-1k (Deng et al., 2009). Using the luminance formula

$$\texttt{gray} = (0.299 \times R) + (0.587 \times G) + (0.114 \times B),$$

we convert RGB images to grayscale and evaluate model accuracy. Along with other color modifications such as swapping various color channels, and including classes other than goldfish and snail, we form the Table 2.

Table 2: Correct ResNet50 predictions out of 50 validation images per class, with various color modifications.

|          | goldfish | snail | ox | broccoli | cucumber | zebra |
|----------|----------|-------|-----|----------|----------|-------|
| original | 48       | 45    | 32  | 46       | 36       | 48    |
| grayscale | 34      | 44    | 29  | 44       | 28       | 48    |
| rg_swap  | 40       | 40    | 26  | 40       | 25       | 49    |
| rb_swap  | 31       | 42    | 28  | 42       | 36       | 48    |
| gb_swap  | 46       | 41    | 26  | 37       | 31       | 49    |

For goldfish, keeping the red channel intact while swapping green and blue does not substantially reduce accuracy, reinforcing the role of the orange color component.

Also note that none of the modifications we use alter the black and white colors. The performance of the model on zebra images is completely unaffected by these changes, suggesting that it is these colors (and not, for example, the green grass on which the zebras might stand) that the model relies on to assign the class label zebra.