# OpenReview forum: "Generating Samples to Probe Trained Models"
_ICLR.cc/2026/Conference — Submitted to ICLR 2026_

### Official Review · Reviewer_pffq · 2025-10-30

**Soundness:** 3
**Presentation:** 3
**Contribution:** 3
**Rating:** 6
**Confidence:** 3

**Summary:**

This paper introduces a framework for generating synthetic data that probes already trained models under various objectives.
It examines four key objectives: uncertainty — data points near the decision boundary; disagreement — where two models make opposite predictions; sensitivity — where small parameter changes strongly affect the output, and counterfactuals —inputs that yield a fixed target label. For each objective they define a probe energy over inputs that induces a Gibbs distribution; sampling from it yields examples for that objective.
The framework is evaluated across toy tasks, real-world tabular datasets, MNIST, and latent probes of pretrained ImageNet classifiers, showing that the generated samples accurately capture each probing objective.

**Strengths:**

The paper is original in framing model probing as a data generation problem using a Bayesian Gibbs formulation. It is clearly written, with solid theoretical grounding and well-chosen illustrative experiments. The framework is significant in that it offers a unified and general-purpose approach for probing trained models —both differentiable and non-differentiable— across multiple axes of behavior, including uncertainty, disagreement, sensitivity, and counterfactual exploration.
It also has potential to extend beyond probing — for example, to adversarial testing, fairness auditing and enforcement, or post-hoc robustness evaluation.

**Weaknesses:**

- Evaluation is mostly qualitative relying primarily on visual examples and descriptive comparisons; adding quantitative metrics for uncertainty, sensitivity,  disagreement and counterfactual would strengthen the evidence.
- The latent-space sampling relies on a pretrained VAE but lacks explicit regularization to prevent drift off the data manifold, which may affect sample fidelity in practice.

- Hyperparameter sensitivity is unexplored; the temperature $\tau$ and sampling parameters likely affect the generated samples. An ablation could clarify the framework’s robustness to these choices.

- No information is provided on runtime or computational cost, which would help assess the practical feasibility of the approach.

- Despite claiming applicability to both classification and regression, all empirical results focus on classification tasks. A regression case study would help substantiate the framework’s generality.


Formulation inconsistencies:
- Although the framework is presented as general over $\mathcal{Y}$, Eq.~(6) for parameter-sensitive samples assumes a binary output through the term $1 - y_{\theta^*}(x)$. The paper should either restrict this objective to binary tasks or provide a general formulation.
- The formulation R_G(x) = $|x - x_a|_r^r$​ in Equations (4)–(7) lacks an explicit regularization weight $\lambda$, which is essential for controlling the trade-off between locality and the probing objective. Including $\lambda$ would improve the completeness and clarity of the formulation.

**Questions:**

1) Why are quantitative metrics for uncertainty, disagreement, sensitivity or counterfactuals not used more broadly to support the probing scenarios?

2) How do you ensure that samples generated in latent space remain on the data manifold, especially without explicit regularization or constraints on the latent variables?

3) How sensitive are the generated samples to the specific design of the probing function G(x) (e.g., choice of loss term, temperature, or regularizer)? Some analysis or ablation could clarify how robust the framework is to these design choices.
Do the experiments include a tunable regularization weight \lambda for R_G(x)?

4) In Figure~6, the gender distribution of the generated counterfactual samples appears roughly balanced, yet the text claims that most samples shift from female to male. Could the authors clarify how this distribution was computed and whether the claimed gender shift is quantitatively supported? Could this observation simply reflect that gender is not a significant factor in the model’s prediction?

5) Eq.~(6) for parameter-sensitive samples appears to assume binary outputs through the term $1 - y_{\theta^*}(x)$. How would this objective be defined for multi-class or regression tasks where such a complement is not well-defined?

---

> ### Author Response · Authors · 2025-11-21
>
> We thank the reviewer for the thoughtful comments.
> Regarding the regression experiment, we did run it. Due to space limitations this could not be included in the main text, but appears in Appendix D (previously Figure 8, now Figure 9), and is now referenced in line 284.
>
> As for the comment on computational cost, all of our experiments were completed on a personal computer within minutes. Thank you for raising this. We now include a brief remark in the manuscript to clarify it (Appendix B)
>
> Q1: We strived for supporting our probing scenarios with quantitative metrics, and summary statistics of the generated data: For model contrasting experiments, we report that the generated data led to opposite predictions in the two models in 100% of the cases (line 278). In the prediction-risky setting, line 1019 of the appendix quantifies how close the generated FICO samples are to the decision boundary. In the wine dataset experiment, line 354 presents the uncertainty levels of the generated samples. For the fixed-label scenario on the US Census dataset, all generated samples were indeed classified as earning more than $50K (flipped decision). Thanks to the referee’s question, we realized that we did not explicitly report this complete success of our approach. We have added this information to the revised version of our paper (line 422).
>
> Q2: The VAE decoder is itself encourages generated samples to remain on the data manifold since the latent space has a smaller dimension than the data space, and a randomly sampled latent vector z followed by the VAE decoder is “more like” a data point than one randomly selected from the pixel space. However, as the reviewer correctly points out, this alone is insufficient for high-dimensional settings such as ImageNet that we presented. In that case, ensuring global plausibility was a technical challenge, since the pretrained VAE only ensured local patch coherence but this was still rich enough that non-plausible images could still be classified by ResNet50 classifier confidently as one given class. As mentioned in line 466, we first generated data samples at a low resolution (the pretrained autoencoder allowed for variable image size, since it was comprised of CNN’s and variable size pooling layers), then progressively increased the resolution as we ran the MALA algorithm.
>
> Q3: Outputs are sensitive to the design of the probing function $G$, since $G$ determines the relative importance of proximity to an anchor point throught the strength of the regularizer. Furthermore, the temperature parameter balances the exploration–exploitation tradeoff in the MALA search. Our experiments do involve both tunable parameters. The referee is right that our presentation did not reflect that, and now we explicitly write the tunable regularizer strength lambda. A practitioner who wishes to emphasize exploration rather than minimizing $G$ can choose a higher temperature or deescalate localization by lowering the regularizer strength parameter. In Appendix C of our revised version, we now include new plots (Figure 8) that illustrate the effect of temperature in the MALA algorithm and the role of the regularization weight lambda.
>
> Q4: In the counterfactual example, the probing function explicitly includes a term that encourages staying close to the original data point, who is a female. If gender were not influential for the model’s prediction, we would expect the generated counterfactuals to remain female, because the regularizer would dominate once the target label is satisfied. The observed shift therefore reflects the model’s dependence on this feature. We have clarified this reasoning in the text (line 429- 431).
>
> Q5: We thank the referee for pointing this out. The same comment also applies to equations (5) and (6), since they both have been formulated only for binary outputs. Upon various rewrites and attempts to adhere to the page limit, the nonbinary formulation of the probing functions had been omitted and a trace remains as a passing comment on line 894 in the appendix. For regression models, one probing function formulation that behaves as desired is $\ell_G = \exp(-\frac{\| y_{\theta*}(x) - y_{\theta}(x) \|^2 }{ \sigma} )$ where $\sigma$ determines the magnitude for what constitutes “differing prediction” for the regression model. The function $G$ (integral of $\ell_G$ over $\theta \sim {q*}(\theta) d\theta$) is high when the two models are aligned and is low as soon as the predictions are larger than \sigma apart. For general categorical output, if we let $y_{\theta}(x)$ be the (post-softmax) probability output of the model, then the same type of exponential would prefer differing outputs in the probability simplex. This type of $\ell_G$ also works for nonbinary model contrast probing function, but now $y_\theta$ and $y_{\theta^*}$ are replaced with $y_1$ and $y_2$ corresponding to the two models. We have added these observations to the revised version of our paper (lines 231-235).

---

### Official Review · Reviewer_FUym · 2025-11-01

**Soundness:** 2
**Presentation:** 2
**Contribution:** 2
**Rating:** 4
**Confidence:** 1

**Summary:**

The paper proposes a general mathematical framework for probing trained models by generating synthetic samples that satisfy custom-defined probing objectives. The approach draws a symmetry between model training (parameter optimization) and model probing (data optimization), and includes analytical and empirical demonstrations across regression and classification tasks.

**Strengths:**

The idea of formulating probing as a generative process is interesting and connects interpretability with probabilistic modeling. Overall, the proposed framework is flexible, allowing different “questions” to be posed to a trained model.

**Weaknesses:**

Overall, I find the paper’s organization difficult to follow, particularly in Section 2. The core mathematical framework is presented in a dense and abstract way, which obscures how the proposed method is actually implemented. Several key equations (e.g., Eq. 2–4) are introduced without sufficient intuition or explanation. Moreover, the design rationale behind the probing function G across different use cases remains unclear—for instance, it is not evident why Eq. (5) appropriately captures model-contrasting samples. These gaps in exposition make it challenging to fully assess the technical quality and contribution of the work.


Minor:
1. Line 84: "First, our notation:" sounds like an unfinished sentence
2. Line 98: "The loss function F in 1" should be "... Equation (1)"

**Questions:**

Can you clarify the computational steps for constructing $G(x)$ and sampling  $p(x)$ in non-trivial models?

---

> ### Author Response · Authors · 2025-11-21
>
> We thank the reviewer for the comments. Equations (2) and (3) are Bayesian objectives in which, rather than minimizing a function $F$ (or $G$) we find a distribution over the space $\Theta$ (or $X$) that balances having its distributional mass focused on points where $F$ is low (or $G$ is low) and also a wide enough distribution. This objective is also called the Evidence Lower BOund (ELBO). Its connection to Bayesian updating, interpreted as optimal information processing, was established by Zellner (1988) in “Optimal Information Processing and Bayes’s Theorem”. In addition to the explanation provided on lines 127-137, we have added a more detailed discussion of this interpretation in Appendix C together with this historical reference.
>
> The choice of $G$ in equation (5) follows directly from the probing question in the model-contrasting scenario: whenever a data point $x$ leads the two models to assign opposing predictions, the function $G$ is low. This means that following a method of producing data samples from the Gibbs distribution (with density $e^{-G(x)}$), we get data samples where the two models have contrasting predictions.
>
> In this paper, we use the efficient Metropolis Adjusted Langevin Algorithm (MALA) for sampling from the distribution. The mathematical details of the method is given in Appendix C. At an intuitive level, each Langevin step of MALA is like a noisy gradient descent move, trying to get to low-loss regions of the landscape (with a precisely added noise term). The Metropolis adjusted part of the algorithm is an acceptance-rejection step that ensures that the limiting distribution of the process indeed samples from the Gibb’s distribution which solves equations (2) and (3). Importantly, our framework is not tied to MALA. Any method that generates samples that balance the objectives in (2) and (3) is suitable. Alternatives include Picard's algorithm or approximate approaches such as variational inference. The specific sampling algorithm is therefore a practical choice rather than a requirement of the framework.

---

### Official Review · Reviewer_skCR · 2025-11-05

**Soundness:** 3
**Presentation:** 3
**Contribution:** 2
**Rating:** 4
**Confidence:** 3

**Summary:**

This paper proposes a general mathematical framework for generating synthetic samples that reflect specific model behaviors. The idea is to take a "dual" view of model training: given a fixed model (or distribution of model parameters), sample data according to a specific user-defined probing function. Samples can be drawn using Langevin dynamics. By choosing different forms of G, the approach can produce various kinds of synthetic samples, such as model-contrastive, prediction-risky, or parameter-sensitive. The method is demonstrated on linear regression, tabular data, and vision models.

**Strengths:**

The paper introduces a clear variational formulation that symmetrically parallels model training and data synthesis. The proposed framework is very general, can handle various kinds of objective functions, and is model-agnostic.

**Weaknesses:**

While the conceptual insight is interesting, at least based on my knowledge in this domain, the paper has relatively marginal novelty in its specific algorithm (which is not necessarily a weakness, though).

The paper currently lacks a discussion of how the proposed framework can be extended to the discrete input space. While a VAE decoder can enforce data-manifold constraints, this is still difficult for generating language data.

The experiments are mostly qualitative and small-scale.

No baselines are being compared in the experiments. However, I am not sure whether there are no prior works on this problem. For example, adversarial examples are being generated using similar gradient-based algorithms.

**Questions:**

See weakness.

---

> ### Author Response · Authors · 2025-11-21
> **Response to questions**
>
> We thank the reviewer for their comments and address the points raised.
>
> Q1: It is true that the main focus of our study is the conceptual insight that model probing can be formulated symmetrically to model training: using loss functions on the data space to fit a fixed trained parameter (or parameter distribution) in direct analogy to using loss functions on the parameter space to fit a model to a fixed data distribution. As the reviewer correctly notes, our goal is not to introduce a new algorithm but to present a general framework for probing models by generating data. The overarching framework idea is independent of the particular algorithm used to generate the data. The framework is deliberately model-agnostic and can incorporate many different probing objectives, while the specific gradient-based algorithm that we use is an efficient and practical instantiation.
>
> Q2: It is true that gradient-based methods do not directly operate on discrete input spaces. However, this does not immediately limit the application of the proposed framework. For example, standard training of and inference with transformers start with embedding discrete language into continuous vectors, and thus gradient based algorithms can be applied in that continuous space. Additionally, in other modalities, there may be relaxations of the discrete input restrictions that might allow gradient based methods to be used. However, the referee does have an intriguing comment.There could indeed be interesting probing questions for language models, for example identifying inputs that trigger jailbreak behaviours or lead the model to produce restricted outputs. Such investigations would require substantial technical development and could form a research project on their own. We will add such a project in our future research agenda.
>
> Q3: Our results are indeed primarily qualitative, as the objective of this first paper is to illustrate the types of insights the framework can reveal. For example, our study of the ImageNet-1k classifier (ResNet-50) shows how probing can expose decision criteria such as color dependence in the “goldfish” class, an insight that we then validated. At the same time, this same experiment demonstrates that not all our settings are small-scale, neither in the dimension of the data-space, nor in the model size.
>
> Q4: One of the many probing scenarios in our general framework is counterfactual data generation and indeed there are different counterfactual generation methods which can be seen as complementary to our particular probing scenario. For example, the well-known and frequently-used approach, DiCE (DIverse Counterfactual Examples) is a method of generating a predetermined number of counterfactual examples that are distinct from each other (with respect to a diversity metric). Unlike our probing scenario, DICE does not generate full probability distributions over inputs aligned with a model’s behaviour. Like DICE and other counterfactual approaches in the literature,  our probing scenario is also  providing  valuable insights into the trained model. To highlight the contrastive and complementary value of our approach, we have conducted a computational comparison of the counterfactual probing scenario against DiCE. A concise discussion is provided in the main text (lines 454-464), with results and discussions presented in Appendix D. We will update our GitHub repository accordingly.

---

### Author Response · Authors · 2025-11-28

We hope that we have addressed the questions posed by the reviewers, and if so we kindly ask of the reviewers to update their ratings to reflect this. We would welcome a discussion if they have further questions.

---

### Comment · Area_Chair_fcBD · 2025-11-29

Dear Reviewers,

Authors’ kindly tried to address your concerns. If the responses address your concerns please acknowledge that. If not, please express remaining concerns. Thanks for your efforts!

Best, AC

---

### Author Response · Authors · 2025-12-02

Dear AC,

Due to the recent ICLR incident, since the referees are not going to be able to respond anymore, we would like to give a summary of the referee reports and our responses for your consideration.

Our submission introduces a framework for understanding the behavior of trained models using loss functions G on the data space, constructed from trained networks model parameters and which are designed to be low when a certain property (of our own choosing) is satisfied. We then use the Variational formulation and sample from the resulting Gibbs-Boltzmann distribution on the data space. The data distribution obtained constitute as answers/explanations to the problems posed by G.

Reviewer pffq finds our framing original, examples well-chosen and clearly written. The reviewer raised certain valid technical points, such as the writing down certain G functions only in the binary classification case, or forgetting to explicitly write the regularization strength parameter. We agree and updated our manuscript to reflect these changes. As per their question on the effect of various hyperparameters (temperature, localizer strength) we also added a new figure (Figure 8 in the appendix) as to clarify it for the reader. In regards to a regression example, the reviewer probably missed Figure 9 and we added a reference to it in the main text.

Reviewer FUym acknowledges the flexibility of our approach, but also in their confidence rating clearly state that they were "unable to assess the paper". We did respond to their technical questions, and changed the wording in the manuscript at certain places to give the intuitive meaning of our sampling procedure, for readers without a background in the variational setup.

Reviewer skCR states that our framework is very general and model agnostic and that our variational formulation is clear. They mention discrete input space, in which case indeed our Metropolis Adjusted Langevin Algorithm (or a Random Walk version) could not be applied. However there seems to be some confusion on the part of the reviewer, as they later mention language inputs. Tokenization and continuous embedding of the language modality allow differentiable training LLMs, and in the same manner the discrete nature of the data would not preclude using our sampling algorithm. The reviewer also mentions that our experiments are qualitative and small scale. The insights we get from the sampling method are indeed from qualitative examination of the data. As for being small scale, we point out our larger examples where we probe a ResNet model trained on Imagenet. Finally the reviewer also asks for comparisons of other data generating methods. We do not have a baseline method to compare to in many of our probing scenarios (one strength of our method is that our G-functions are general). But indeed in certain cases, such as generating counterfactual scenarios, we did follow through with the request of the reviewer and included a computational comparison of the counterfactual probing scenario against DiCE, which ends up yielding complementary results to those we produced.

---

### Meta-Review · Area_Chair_fcBD · 2026-01-05

**Summary:**

The paper proposes a theoretical framework that treats model probing as a "dual" to model training: instead of optimizing parameters for fixed data, it optimizes data input to satisfy specific loss functions $G$ (probing objectives) for a fixed model. This is achieved via a Variational formulation and sampling using the Metropolis Adjusted Langevin Algorithm (MALA).The reviewers generally appreciated the conceptual elegance, symmetry, and model-agnostic nature of the framework. However, initial concerns were raised regarding the novelty of the sampling algorithm, the lack of quantitative baselines, the applicability to discrete domains (e.g., text), and the clarity of the mathematical formulation for non-experts.

**Reviewer Concerns:**

Resolved Concerns:

Baselines and Comparisons (Reviewer skCR): The reviewer noted a lack of baselines. The authors clarified that for many probing scenarios, no direct baseline exists, but they successfully added a computational comparison against DiCE (Diverse Counterfactual Examples) for the counterfactual scenario.

Regression Examples (Reviewer pffq): The reviewer requested non-classification examples. The authors pointed to existing regression results in the Appendix (Figure 9) which the reviewer had initially missed.

Outstanding / Partially Resolved Concerns:

Discrete Input Spaces (Reviewer skCR): The reviewer questioned the applicability to language models. While the authors argued that gradient-based methods apply to continuous embedding spaces, they acknowledged that addressing discrete token limitations and "jailbreak" style probing would require substantial future technical development.

Quantitative Rigor (Reviewer skCR, pffq): Both reviewers noted the reliance on qualitative/visual inspection. While the authors highlighted specific quantitative stats (e.g., 100% flip rate in counterfactuals), the primary value of the paper remains in qualitative insight generation rather than beating SOTA metrics on a leaderboard.

Mathematical Accessibility (Reviewer FUym): This reviewer found the connection between the ELBO and the probing objective opaque. The authors provided the derivation and historical context (Zellner, 1988) in the revision. However, given the reviewer's self-admitted low confidence, it is unclear if the revision fully bridged this gap for a general audience.

**Reviewer Scores:**

No change

---

### Decision · Program_Chairs · 2026-01-26

Reject